# AgentNoiseBench: Benchmarking Robustness of Tool-Using LLM Agents Under Noisy Condition

**Ruipeng Wang**[1][*] **Yuxin Chen**[2][3][*] **Yukai Wang**[1][*] **Chang Wu**[1][*] **Junfeng Fang**[3] **Xiaodong Cai**[2] **Qi Gu**[2][†] **Hui Su**[2] **An Zhang**[1] **Xiang Wang**[1][†] **Xunliang Cai**[2] **Tat-Seng Chua**[3]

## Abstract

Recent advances in large language models have enabled LLM-based agents to achieve strong performance on a variety of benchmarks. However, their performance in real-world deployments often that observed on benchmark settings, especially in complex and imperfect environments. This discrepancy largely arises because prevailing training and evaluation paradigms are typically built on idealized assumptions, overlooking the inherent stochasticity and noise present in real-world interactions. To bridge this gap, we introduce AgentNoiseBench, a framework for systematically evaluating the robustness of agentic models under noisy environments. We first conduct an in-depth analysis of biases and uncertainties in real-world scenarios and categorize environmental noise into two primary types: usernoise and tool-noise. Building on this analysis, we develop an automated pipeline that injects controllable noise into existing agent-centric benchmarks while preserving task solvability. Leveraging this pipeline, we perform extensive evaluations across a wide range of models with diverse architectures and parameter scales. Our results reveal consistent performance variations under different noise conditions, highlighting the sensitivity of current agentic models to realistic environmental perturbations. Code is available at https://github.com/keven-cyber/agentnoisebench

## 1. Introduction

Recent advances in large language models (LLMs) (OpenAI, 2025d; Google, 2025c; Team et al., 2025b) have substantially improved their reasoning and tool-use capabilities (Team et al., 2025a; Zeng et al., 2025a; Team et al., 2025c), enabling the development of increasingly capable LLM-based agents. As a result, agentic models have demonstrated strong performance across a wide and growing range of benchmarks (Yao et al., 2024; Barres et al., 2025; He et al., 2025). However, recent studies indicate that many agents experience notable performance degradation when encountering out-of-distribution settings or operating in complex real-world environments (Deng et al., 2023; Zhou et al., 2023; Xue et al., 2025).

We argue that a fundamental gap exists between the assumptions underlying current agent evaluations and the conditions encountered in realistic environments. In particular, most existing benchmarks evaluate agents under idealized assumptions, where instructions are carefully curated, and interactions with the environment are stable and well-controlled (Zeng et al., 2024; Qi et al., 2024; Jin et al., 2025). In contrast, real-world environments are inherently stochastic and imperfect. User interactions exhibit substantial diversity and unpredictability (Gallois et al., 2005; Trippas et al., 2024; Wang et al., 2024), while external tools frequently return noisy, incomplete, or failed outputs due to uncontrollable factors (Vuddanti et al., 2025; Xiong et al., 2025). Despite the prevalence of such noise in practice, existing benchmarks often fail to systematically expose agents to these factors, making robustness under realistic conditions difficult to assess.

Systematically evaluating agent robustness under noise remains non-trivial. Concretely, there are several main challenges: (1) Absence of a systematic taxonomy: agents are exposed to diverse and heterogeneous noise sources in realistic environments, yet existing benchmarks provide no unified framework for characterizing them. (2) Complexity of noise injection: injecting realistic noise in a principled manner is non-trivial, as it requires balancing increased environmental complexity with the preservation of task solvability. (3) Limitations of evaluation protocols: evaluating agent

---

*Equal contribution. [1]University of Science and Technology of China, China [2]Meituan, China [3]National University of Singapore, Singapore. Correspondence to: Xiang Wang <xiangwang1223@gmail.com>, Qi Gu <guqi03@meituan.com>.

*Proceedings of the $43^{rd}$ International Conference on Machine Learning*, Seoul, South Korea. PMLR 306, 2026. Copyright 2026 by the author(s).

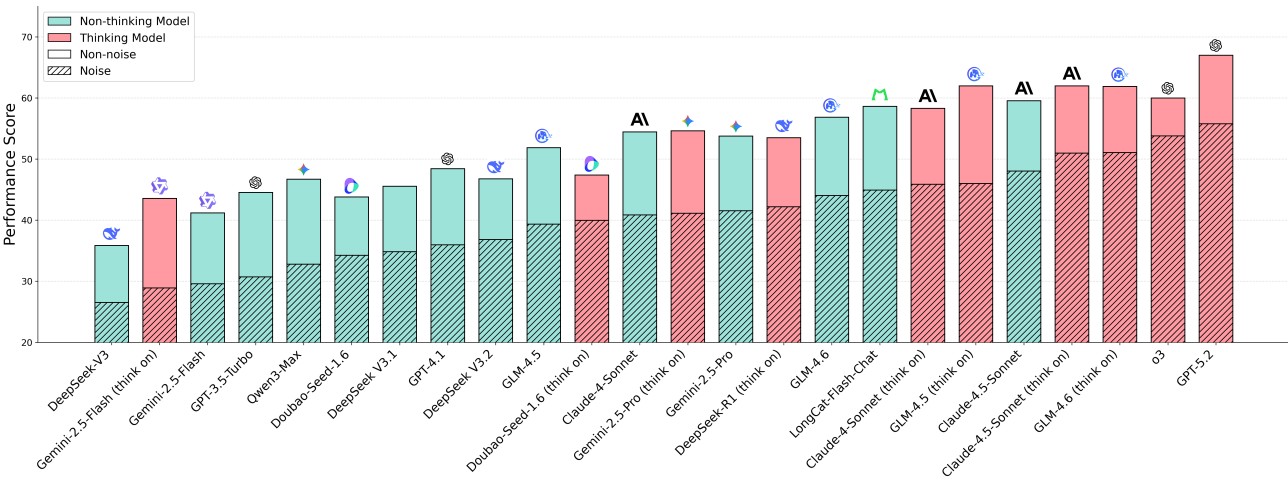

*Figure 1.* Overall scores on AgentNoiseBench, sorted by model performance under noisy conditions.

behavior under noise is inherently multi-faceted, while current benchmarks lack comprehensive evaluation protocols to capture such behaviors.

To this end, we introduce AgentNoiseBench, a framework designed to evaluate the noise robustness of agentic models. Specifically, we conduct a systematic analysis of biases observed in real-world environments and identify two primary sources of noise: (1) user-noise, which captures the inherent ambiguity and variability in user interaction patterns, and (2) tool-noise, which simulates execution failures, inconsistent responses, and partial results from external tools. Building on this analysis, we develop an automated pipeline to inject both noise types into existing agentic benchmarks, while enforcing strict constraints to preserve task solvability. Furthermore, we introduce a trajectory-aware evaluation protocol that enables multi-dimensional analysis of agent behavior under noisy conditions.

We instantiate AgentNoiseBench across two representative agentic capabilities—tool use and search—by injecting controlled user-noise and tool-noise into $\tau^2$-Bench and VitaBench (Barres et al., 2025; He et al., 2025), as well as HotPot QA (Yang et al., 2018). We then evaluate a broad set of open-source and proprietary models, including both reasoning-oriented models and standard foundation models, spanning diverse architectures and parameter scales. As shown in Figure 1, all evaluated models exhibit consistent performance degradation under both noise sources, with varying degrees of sensitivity. Further analysis indicates that general reasoning ability and environmental robustness are not strongly correlated. By examining the step-wise entropy of agent trajectories, we find that different noise sources affect the entropy minimization process through distinct mechanisms. Finally, systematic failure pattern analysis shows that models across all families are more sensitive to

tool-noise than to user-noise, potentially due to the inherent variability and limited controllability of the user simulator.

Overall, the main contributions of this work are:

- We conduct a systematic analysis of biases in realistic environments and categorize them into two primary noise sources: user-noise and tool-noise.

- We develop an automated pipeline that injects controllable noise into existing agentic benchmarks while preserving task solvability.

- We introduce a trajectory-aware evaluation protocol and perform extensive evaluations across a wide range of open-source and proprietary models, yielding key insights into agent robustness.

**Conflict of Interest Disclosure**   The authors Y. Chen, X. D. Cai, Q. Gu, H. Su, X. L. Cai are employed by Meituan, which leads the development of LongCat-Flash-Chat, one of the models evaluated in this work.

## 2. AgentNoiseBench: A Systematic Framework for Robustness Evaluation

In this section, we formally present **AgentNoiseBench**, a systematic evaluation framework designed to quantify the robustness of LLM-based agents against both random perturbations (Song et al., 2024; Li et al., 2024) and adversarial imperfections (Zhan et al., 2024; Zhang et al., 2025) inherent in real-world deployment.

### 2.1. Design Principles

Real-world environments are inherently imperfect (Wang et al., 2025a; Siska et al., 2024; Lunardi et al., 2025). To

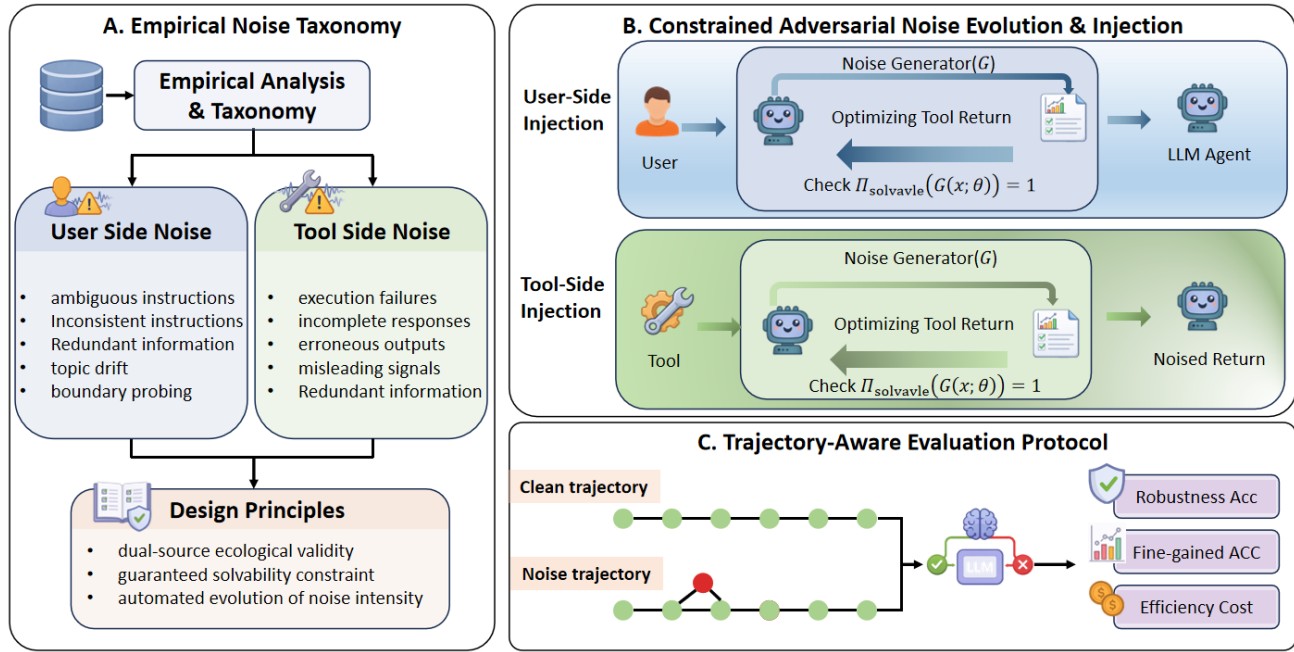

*Figure 2.* The framework of AgentNoiseBench. **(A) Empirical Noise Taxonomy** categorizes real-world noise into instruction noise and tool-execution noise. **(B) Constrained Adversarial Noise Evolution & Injection** mechanism applies controlled noise while ensuring task solvability. **(C) Trajectory-Aware Evaluation Protocol** assesses agent behavior and robustness through multi-dimensional metrics.

address this evaluation gap, AgentNoiseBench bridges the gap between idealized benchmark evaluations and the complexities of realistic environments through three governing principles:

**Realistic Noise Modeling.** Rather than injecting arbitrary noise, we model valid noise grounded in real-world interaction logs. Specifically, we categorize noise into user-noise, reflecting user cognitive biases and linguistic variability (Zhang et al., 2024a; Gan et al., 2024; Rivkin et al., 2023), and tool-noise, simulating instability and unreliability in external environments (Xu et al., 2024a; Kokane et al.; Chen et al., 2025; Zhang et al., 2024b).

**Solvability-Preserving Noise Injection.** A core premise of our framework is that injected noise should increase task difficulty without rendering the task unsolvable. Accordingly, all perturbations are constrained to preserve the necessary conditions for a valid solution, ensuring that failures can be attributed to agent fragility rather than task invalidity.

**Realistic Evaluation Requirements.** Outcome-based metrics are insufficient as they fail to distinguish between robust reasoning and stochastic luck. Therefore, a rigorous evaluation must demand process-level correctness: the agent's intermediate reasoning steps should remain logically sound and strictly adhere to task requirements, impervious to noise-induced deviations. This ensures that the evaluation metrics faithfully reflect the agent's robustness to noise.

## 2.2. Benchmark Construction

To enable fair comparisons across agents, AgentNoiseBench adopts a standardized benchmark construction pipeline. Concretely, we first derive realistic noise types from large-scale user–agent interaction logs. We then instantiate a fixed adversarial noise generator to produce challenging yet valid perturbations, applying it uniformly to all agents. This design keeps the noise distribution and difficulty level consistent across models, thereby enabling controlled and fair evaluations.

### 2.2.1. EMPIRICAL TAXONOMY OF NOISE

We categorize noise types through a systematic analysis of large-scale user–agent interaction logs. Concretely, we combine manual annotation with frequency-based statistical analysis to identify noise types that are both prevalent in real-world interactions and associated with high interaction costs. We group the resulting noise into two broad families: user noise and tool noise.

**User Noise ($\mathcal{N}_{\text{user}}$):** Models the imperfections and variability in human-provided instructions during agent interactions. Concretely, we consider the following categories of user-side noise:

- **Ambiguous Instructions:** User instructions that are underspecified or lack explicit constraints.

- **Inconsistent Instructions:** User instructions are contradictory or conflicting across multi-turn interactions.

- **Redundant Information:** User instruction include irrelevant contextual information.

- **Topic Drift:** User instructions that shift the primary task intent or conversational focus across dialogue turns.

- **Boundary Probing:** User instructions that deliberately test the limits of the agent's knowledge.

**Tool Noise ($\mathcal{N}_{\text{tool}}$):** Models the stochasticity and unreliability of external tools, APIs, and environments that the agent interacts with. Concretely, we consider the following categories of tool-side noise:

- **Execution Failures:** Tool invocations fail to return valid responses due to service-level issues, including unavailable or uninitialized tools, inaccessible external services, or server-side exceptions.

- **Incomplete Responses:** Tools return outputs with missing or partially truncated data.

- **Erroneous Outputs:** Tool responses containing incorrect or inconsistent factual information.

- **Misleading Signals:** Tool outputs include misleading or unreliable content that can divert agents toward incorrect reasoning trajectories.

- **Redundant Information:** Tools return excessive irrelevant or low-utility content, unnecessarily increasing context length and processing overhead.

### 2.2.2. CONSTRAINED ADVERSARIAL NOISE INJECTION

To generate noise that is strictly valid yet sufficiently challenging, while ensuring fairness across diverse models, we introduce a constrained adversarial injection mechanism. Specifically, we employ a state-of-the-art model (e.g., o3) as a fixed reference agent $A_{\text{ref}}$ to construct a universal noise generation system prompt.

We instantiate a strong language model as the noise generator $\mathcal{G}$, with its parameters frozen. Optimization is performed exclusively over the adversarial system prompt $\theta$. Given an agent input $x$ at a particular interaction step, which is either a user instruction or a tool return, the generator produces a perturbed input $\mathcal{G}(x; \theta)$. Our objective is to identify an optimal prompt strategy $\theta^*$ that maximally degrades the performance of the reference agent $A_{\text{ref}}$, subject to strict preservation of the underlying task solvability:

$$
\begin{aligned}
\theta^* = \arg\max_{\theta} \ & \mathcal{L}_{\text{deg}}(A_{\text{ref}}, \mathcal{G}(x; \theta)) \\
\text{s.t.} \quad & \mathbb{I}_{\text{solvable}}(\mathcal{G}(x; \theta)) = 1.
\end{aligned} \tag{1}
$$

Here, $\mathcal{L}_{\text{deg}}$ quantifies the performance degradation of the reference agent under injected noise, while $\mathbb{I}_{\text{solvable}}$ enforces

that the perturbed input remains solvable. We iteratively update $\theta$ to maximize $\mathcal{L}_{\text{deg}}$ under the solvability constraint, thereby producing valid yet adversarial noise.

By optimizing the noise prompt against a state-of-the-art reference agent, we leverage the transferability of noise across agents: noise that reliably disrupts a highly capable reference agent tends to generalize and remain challenging for other agents. After obtaining $\theta^*$, we freeze it and apply it uniformly when generating noise for all evaluated agents. This ensures that all agents are evaluated under identical noise conditions, enabling a fair and consistent comparison.

### 2.3. Trajectory-Aware Evaluation Protocol

We retain the standard evaluation metrics from VitaBench (He et al., 2025) and $\tau^2$-Bench (Barres et al., 2025) to assess final outcome correctness. To more comprehensively assess agent behavior, we introduce a Trajectory-Aware Evaluation Protocol. This protocol examines the full interaction trajectory and accounts for cases where a correct final answer is obtained via deviated reasoning or interaction paths.

**Trajectory Deviation Detection.** Let $\tau = (s_1, s_2, \ldots, s_T)$ denote the agent's reasoning trajectory. Given task requirements $\mathcal{T}$, we define a step-wise validity indicator $\mathbb{I}_{\text{step}}$. For each state $s_i \in \tau$, $\mathbb{I}_{\text{step}}(s_i, \mathcal{T})$ assesses whether the agent's behavior at that step exhibits a noise-induced deviation:

$$
\mathbb{I}_{\text{step}}(s_i, \mathcal{T}) = \begin{cases} 1, & s_i \text{ is valid and consistent with } \mathcal{T}, \\ 0, & \text{otherwise.} \end{cases} \tag{2}
$$

A trajectory is deemed valid only if all steps remain consistent. Consequently, the trajectory-level validity is defined as the conjunction of step-wise indicators:

$$
\mathbb{I}_{\text{traj}}(\tau, \mathcal{T}) = \prod_{i=1}^{T} \mathbb{I}_{\text{step}}(s_i, \mathcal{T}). \tag{3}
$$

**Stability-Gated Success Criterion.** We apply trajectory-level validity as a gating condition for task execution success. Under this criterion, a task is considered successfully solved only when the agent produces the correct final outcome and the corresponding reasoning trajectory remains valid throughout the interaction:

$$
\text{SGA}(\tau; \mathcal{T}) = \mathbb{I}_{\text{traj}}(\tau; \mathcal{T}) \cdot \mathbb{I}_{\text{task}}(s_T; \mathcal{T}), \tag{4}
$$

where $\mathbb{I}_{\text{task}}(s_T; \mathcal{T}) \in \{0, 1\}$ indicates whether the final outcome is correct for task $\mathcal{T}$. This criterion filters out cases where a correct final outcome is achieved despite deviations in the interaction trajectory.

*Table 1.* Model performance across different domains. Values are reported as mean±std over multiple runs, where std denotes the standard deviation. For each domain, the best-performing model within the non-thinking and thinking groups is highlighted in **bold**.

| Models | $\tau^2$-bench | | | Vitabench | | | Search | |
|---|---|---|---|---|---|---|---|---|
| | Origin | Tool_Noise | User_Noise | Origin | Tool_Noise | User_Noise | Origin | Tool_Noise |
| **Non-thinking Models** | | | | | | | | |
| DeepSeek-V3-0324 | 0.47±0.02 | 0.31±0.01 | 0.37±0.03 | 0.25±0.02 | 0.17±0.01 | 0.20±0.02 | 0.21±0.01 | 0.22±0.03 |
| Gemini-2.5-Flash (think off) | 0.56±0.04 | 0.31±0.02 | 0.50±0.01 | 0.26±0.03 | 0.16±0.02 | 0.22±0.01 | 0.28±0.02 | 0.20±0.02 |
| Doubao-Seed-1.6 | 0.51±0.03 | 0.31±0.02 | 0.45±0.04 | 0.37±0.01 | 0.28±0.03 | 0.32±0.02 | 0.26±0.03 | 0.20±0.02 |
| GPT-4.1 | 0.56±0.02 | 0.30±0.01 | 0.50±0.03 | 0.41±0.04 | 0.31±0.02 | 0.33±0.01 | 0.29±0.03 | 0.27±0.04 |
| DeepSeek-V3.1 (w/o thinking) | 0.55±0.01 | 0.35±0.03 | 0.45±0.02 | 0.36±0.03 | 0.31±0.01 | 0.30±0.02 | 0.43±0.04 | 0.37±0.01 |
| DeepSeek-V3.2-Exp (w/o thinking) | 0.52±0.03 | 0.34±0.02 | 0.49±0.01 | 0.42±0.04 | 0.32±0.03 | 0.32±0.01 | 0.45±0.02 | 0.36±0.04 |
| gemini-2.5-pro (w/o thinking) | 0.59±0.02 | 0.37±0.04 | 0.55±0.03 | **0.49**±0.01 | 0.34±0.02 | 0.39±0.04 | 0.36±0.03 | 0.29±0.03 |
| gpt-3.5-turbo (w/o thinking) | 0.53±0.04 | 0.28±0.01 | 0.46±0.02 | 0.36±0.03 | 0.23±0.01 | 0.26±0.03 | 0.17±0.02 | 0.13±0.01 |
| Qwen3-Max | 0.57±0.01 | 0.34±0.03 | 0.51±0.04 | 0.36±0.02 | 0.23±0.03 | 0.23±0.02 | 0.31±0.03 | 0.24±0.02 |
| GLM-4.5 (w/o thinking) | 0.62±0.03 | 0.42±0.02 | 0.55±0.01 | 0.41±0.04 | 0.29±0.01 | 0.31±0.03 | 0.44±0.02 | 0.36±0.02 |
| GLM-4.6 (w/o thinking) | **0.74**±0.02 | 0.48±0.04 | 0.64±0.03 | 0.40±0.01 | 0.32±0.02 | 0.33±0.04 | **0.47**±0.01 | **0.39**±0.02 |
| LongCat-Flash-Chat | 0.74±0.02 | 0.48±0.01 | 0.62±0.02 | 0.43±0.02 | 0.34±0.04 | 0.36±0.01 | 0.38±0.03 | 0.30±0.02 |
| Claude-4-Sonnet (w/o thinking) | 0.65±0.01 | 0.41±0.03 | 0.55±0.04 | 0.44±0.02 | 0.34±0.04 | 0.33±0.03 | 0.38±0.04 | 0.30±0.01 |
| Claude-4.5-Sonnet (w/o thinking) | 0.71±0.03 | **0.48**±0.02 | **0.64**±0.01 | 0.48±0.04 | **0.40**±0.03 | **0.39**±0.02 | 0.34±0.01 | 0.29±0.03 |
| **Thinking Models** | | | | | | | | |
| Qwen3-max (w/ thinking) | **0.87**±0.02 | 0.56±0.03 | **0.78**±0.04 | 0.47±0.02 | 0.34±0.01 | 0.40±0.01 | **0.49**±0.03 | 0.33±0.02 |
| Gemini-2.5-Flash (think on) | 0.58±0.01 | 0.31±0.04 | 0.50±0.02 | 0.29±0.03 | 0.14±0.01 | 0.20±0.04 | 0.27±0.04 | 0.22±0.02 |
| Doubao-Seed-1.6-Thinking | 0.63±0.03 | 0.40±0.01 | 0.61±0.03 | 0.31±0.02 | 0.28±0.04 | 0.31±0.01 | 0.27±0.03 | 0.21±0.02 |
| GLM-4.5 (w/ thinking) | 0.70±0.04 | 0.49±0.02 | 0.63±0.01 | 0.47±0.04 | 0.35±0.03 | 0.37±0.02 | 0.45±0.02 | 0.37±0.01 |
| GLM-4.6 (w/ thinking) | 0.75±0.01 | 0.56±0.04 | 0.68±0.02 | 0.49±0.01 | 0.40±0.04 | 0.40±0.03 | 0.47±0.04 | 0.36±0.02 |
| Claude-4-Sonnet (w/ thinking) | 0.69±0.03 | 0.46±0.01 | 0.58±0.04 | 0.43±0.02 | 0.36±0.01 | 0.36±0.04 | 0.39±0.03 | 0.31±0.02 |
| Claude-4.5-Sonnet (w/ thinking) | 0.73±0.02 | 0.57±0.00 | 0.65±0.02 | 0.51±0.03 | 0.41±0.02 | 0.41±0.01 | 0.35±0.02 | 0.29±0.02 |
| deepseek-R1-0528 (w/ thinking) | 0.53±0.01 | 0.37±0.01 | 0.46±0.02 | 0.42±0.04 | 0.39±0.03 | 0.36±0.02 | 0.33±0.03 | 0.26±0.02 |
| gpt-5.2 (w/ thinking) | 0.81±0.04 | **0.61**±0.03 | 0.72±0.03 | **0.53**±0.01 | 0.45±0.04 | **0.45**±0.03 | 0.49±0.03 | **0.37**±0.03 |
| o3 (w/ thinking) | 0.70±0.02 | 0.57±0.01 | 0.68±0.02 | 0.50±0.02 | **0.46**±0.01 | 0.44±0.04 | 0.46±0.03 | 0.35±0.02 |
| Gemini-2.5-Pro | 0.60±0.02 | 0.37±0.03 | 0.55±0.01 | 0.49±0.04 | 0.34±0.02 | 0.38±0.03 | 0.37±0.03 | 0.30±0.03 |

## 3. Experiments

### 3.1. Experimental Setup

**Model Zoo.** We evaluate a diverse set of 24 representative large language models covering different architectures and parameter scales. Our evaluation benchmark encompasses a wide spectrum of proprietary models, including the OpenAI o-series (o3) (OpenAI, 2025e) and GPT series (GPT-4.1 (OpenAI, 2025a), GPT-5.2 (OpenAI, 2025b;c;d)); the DeepSeek family, covering the R1 series (Guo et al., 2025) and V3 variants (V3-0324 (Liu et al., 2024), V3.1 (DeepSeekAI, 2025), V3.2 (Liu et al., 2025)); Anthropic's Claude series (Claude-4-Sonnet (Anthropic, 2025a), Claude-4.5-Sonnet (Anthropic, 2025b)); and Google's Gemini-2.5 suite (Flash and Pro (Comanici et al., 2025; Google, 2025b;a)). Additionally, we include other leading models such as Qwen3-Max (Team, 2025), GLM variants (GLM-4.5 (Zeng et al., 2025b), GLM-4.6 (z.ai, 2025)), Seed-1.6 (ByteDance, 2025), and LongCat-Flash (Team et al., 2025d). To ensure a fair comparison, we distinguish between reasoning-enhanced and non reasoning models. Hybrid architectures capable of toggling between modes are assessed in both states independently.

**Datasets.** To systematically evaluate agent robustness in complex environments, we delineate core capabilities into two dimensions: (1) static knowledge retrieval and (2) dynamic multi-turn interaction. Specifically, for static retrieval tasks, we employ two representative benchmarks: 2Wiki-MultiHopQA (Ho et al., 2020) and HotpotQA (Yang et al., 2018). Collectively, the averaged performance across these two benchmarks is reported as 'Search' , representing static knowledge retrieval. Given that these tasks inherently lack user interaction, tool noise is injected. Conversely, for dynamic interaction tasks, we utilize VitaBench and $tau^2$-Bench, two benchmarks that support the simultaneous injection of both user noise and tool noise. Across all test domains, noise is introduced by perturbing key interaction interfaces, while strictly preserving the invariance of the original task objectives. Finally, to strike a balance between the experimental necessity of covering all predefined noise types and the practical constraints of evaluation scalability and cost, we adopt a stratified sampling strategy: 25% of samples are drawn from each scenario within VitaBench and $au^2$-Bench, and 500 instances are sampled from the multi-hop QA datasets. This design ensures precise comparative analysis of model performance in both noisy and

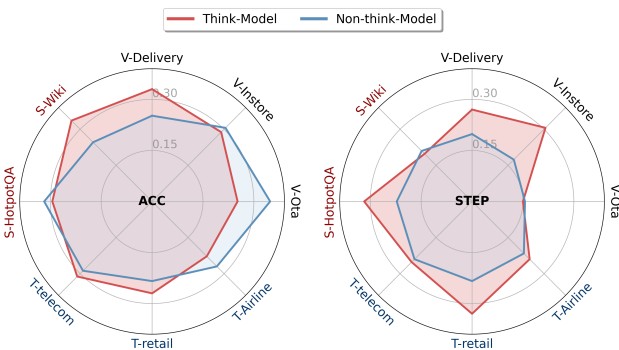

*Figure 3.* Relative deviation metrics including Accuracy (ACC) and Inference Steps (STEP) for Think and Non-Think models across diverse scenarios. For brevity, prefix "V-", "T-" and "S-" stands for VitaBench, $\tau^2$-Bench, and Search.

noise-free environments under a unified task distribution.

**Metrics.** We report mean performance over $N = 4$ independent trials per task. Our evaluation captures both effectiveness and efficiency: Avg@4: The average task success rate across trials, serving as the primary measure of agent robustness. Avg_Tokens@4: The average token consumption across trials, quantifying computational cos Avg_Steps@4: The average number of interaction steps required to complete the task across trials, quantifying computational cost. Robustness: The relative rate of change of a metric above from noise-free to noisy environments, quantifying agent robustness under noisy conditions.

### 3.2. Noise Robustness Remains an Open Challenge for Current Agents

To assess whether noise robustness constitutes a critical bottleneck for agent deployment, we evaluated a series of state-of-the-art models following the experimental settings described in Section 3.1. Table 1 summarizes the performance of each model under both non-noise and noisy environments. Based on the experimental results, we make the following observation:

**Observation 1: Agents universally exhibit significant vulnerability to noise, with varying degrees of sensitivity.** Our results indicate that nearly all models suffer substantial performance degradation upon noise injection, with an average accuracy drop of 20.8%. This performance loss is prevalent across all evaluation domains; for instance, the average declines in the Vitbench, $\tau^2$-bench, and Search domains reach 24.1%, 21.2%, and 16.8%, respectively. Furthermore, we observe significant disparities in the magnitude of performance degradation across different models when subjected to identical noise interference. These results not only reveal the significant performance gap between an ideal noise-free environment and a real noisy scenario, but also highlight the

intrinsic differences in models' sensitivity to noise, raising concerns for the reliable deployment of current agents.

### 3.3. Misalignment Exists between Reasoning Ability and Robustness

To further explore the relationship between general reasoning ability and noise robustness, we separately calculated the robustness scores of thinking models and non-thinking models. The comparative results are visualized in Figure 3, from which we summarize the following finding:

**Observation 2: Reasoning ability and robustness are not closely related.** Our experimental results indicate that reasoning ability and noise robustness are not closely related. Specifically, as shown in Figure 3, with the exception of specific scenarios (e.g., Delivery), the robustness scores of thinking models are relatively lower than those of non-thinking models across most other scenarios. This performance disadvantage prevalent across multiple domains clearly indicates that strong reasoning ability does not inherently confer robustness against noise. To deeply investigate the root causes of the low robustness observed in powerful reasoning models under noisy conditions, we analyzed the models' reasoning trajectories. Our study uncovered an interesting phenomenon: reasoning models often fail to effectively filter out noisy information. Instead, they tend to misinterpret this noise as meaningful, critical signals, and on this basis, attempt to construct complex logical chains to rationalize the noise. This mechanism leads to the generation of spurious reasoning chains—where the model produces solutions that appear logically rigorous but are in fact incorrect. The fundamental reason for this behavior is closely related to the training of reasoning models; their training data is typically strictly screened and pre-processed to eliminate noise. Consequently, when noise appears during the testing phase, the powerful reasoning capability of these models more readily fabricates information to make the noise seem rational.

### 3.4. Impact of Noise Sources and Granularity on Agent Performance

To further investigate the distinct effects of noise on agents, as shown in Figure 4(b) and Figure 4(a), we analyze noise from both a macro perspective of noise sources and a micro perspective of fine-grained noise types. Based on the experimental results, we summarize the following observations.

**Observation 3: Agents exhibit higher sensitivity to tool-side noise.** The results reveal a substantial robustness gap when agents are exposed to different noise sources: across model families and scenarios, agents are significantly more sensitive to tool noise than to user noise. Quantitative analysis shows that tool noise typically results in severe performance degradation of 20To explain this asymmetric sensi-

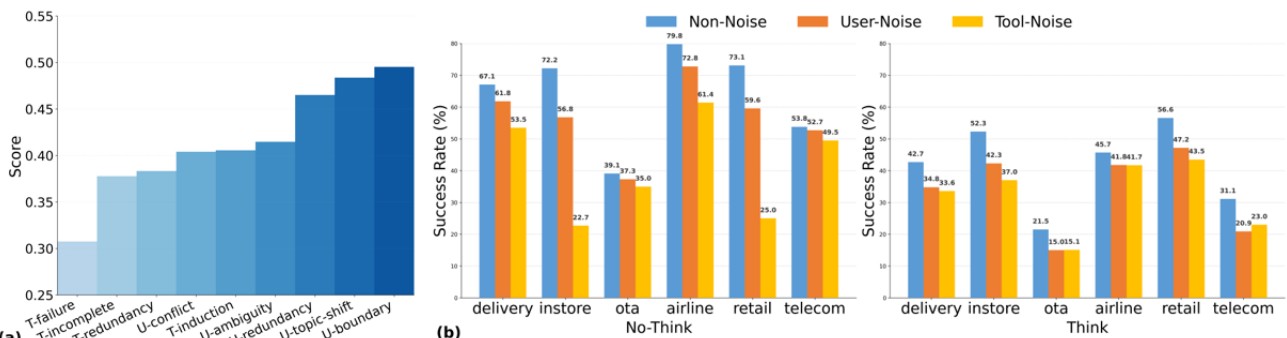

*Figure 4.* The impact of different noises on agent performance. (a) Average impact of nine fine-grained noise categories (user-side "U-" and tool-side "T-") across all scenarios. (b) Performance degradation caused by user noise vs. tool noise in non-reasoning and reasoning-enabled models.

tivity, we analyze the agent interaction logic and attribute it to differences in the nature of user noise and tool noise.

User noise exhibits semantic repairability. User noise is essentially a perturbation at the level of language expression. During pretraining, large language models acquire extensive prior knowledge, which endows them with strong error correction and associative capabilities. As a result, even when input instructions are noisy, agents can often leverage internal commonsense knowledge to repair such noise at the semantic level, thereby maintaining normal task execution.

In contrast, tool noise leads to a complete loss of reasoning evidence. Tool-side noise directly corrupts the agent's channel for perceiving the external environment. During task execution, tool outputs serve as the only objective basis for assessing the current state and deciding subsequent actions. The absence or corruption of such objective facts constitutes a structural failure: agents cannot infer the environment state from internal reasoning, causing the reasoning chain to lose its foundation and ultimately leading to task failure.

**Observation 4: Fine-grained noise types exhibit significantly different destructive effects on agents.** The experimental results indicate that the impact of fine-grained noise on agents' performance is highly uneven. On the user side, instruction contradiction is the most destructive noise type. Agents struggle to infer a consistent and correct intent from logically incompatible instructions, resulting in a pronounced performance drop. In comparison, instruction ambiguity leads to only limited performance degradation. This suggests that although ambiguous intent increases interpretation difficulty, agents can often rely on probabilistic intent inference and partial semantic repair, supported by the rich prior knowledge of large language models, thereby partially mitigating the negative impact of noise. On the tool side, execution failures are the most destructive, reducing the average performance score to 0.31. A reasonable explanation is that such noise triggers structural interruptions in execution, cutting off the agent's access to environmental

feedback and preventing the reasoning process from progressing due to the lack of external factual grounding. By contrast, noise such as incomplete information or redundancy have relatively milder effects, suggesting that agents retain some ability to filter and complete information, extracting useful signals from noisy feedback to sustain correct reasoning processes.

## 3.5. When Does Noise Injection Have the Greatest Impact

To examine the impact of noise injection timing on agent performance, we divide the task execution trajectory into three noise injection stages: early, middle, and late. We conduct experiments on four representative model families, Claude, Gemini, DeepSeek, and LongCat. The experimental results are shown in Figure 5, with additional results for more models provided in Appendix A.2. Based on these results, we summarize the following observation:

**Observation 5: Agents are most sensitive to noise injected at the middle stage.** Specifically, for most evaluated models, noise injected at the middle stage leads to greater performance degradation than noise injected at either the early or late stage, indicating that agents are more sensitive to perturbations occurring in the middle of the execution trajectory. One possible explanation is that, at the early stage, the model maintains higher uncertainty about the environment and thus reasons more conservatively, while at the late stage the state space has largely contracted and action choices are more constrained. In contrast, during the middle stage, the model tends to make more deterministic decisions based on accumulated contextual information, causing injected noise to be more easily absorbed as reliable signals. Notably, this phenomenon is highly consistent across models with different architectures and reasoning strategies, suggesting that heightened sensitivity to middle-stage noise is likely an inherent property of the agent's execution process rather than a model-specific phenomenon.

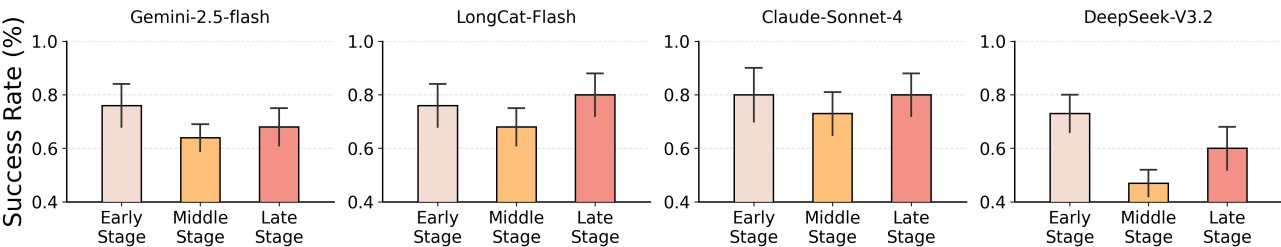

*Figure 5.* Impact of noise injection stages on model performance across different architectures.

## 4. Related Work

### 4.1. Benchmarks for Tool-Using and Interactive Agents

Recent advances have significantly expanded the scope of agent evaluation from single-turn tasks (Huang et al., 2023; Qin et al., 2023; Patil et al.) to complex reasoning scenarios involving tool use and multi-step decision making (Farn & Shin, 2023; Wang et al., 2023; Qian et al., 2024). A growing body of work focuses on agents that iteratively interact with external tools to solve problems (Lu et al., 2025; Wang et al., 2025b; Qian et al., 2025), establishing protocols to assess planning capabilities (Yao et al., 2022; Schick et al., 2023; Zhou et al., 2023). Furthermore, benchmarks like $\tau^2$-Bench (Barres et al., 2025) and VitaBench (He et al., 2025) have been developed to evaluate agents in multi-turn interactions where maintaining state is crucial.

However, a significant gap remains between benchmark environments and real-world applications. Most existing frameworks rely on idealized assumptions (Yang et al., 2024; Wang et al., 2025d), providing agents with carefully curated instructions and well-controlled environments. Unlike prior work, AgentNoiseBench introduces an automated pipeline capable of injecting noise into any existing agent-centric benchmarks while maintaining task solvability.

### 4.2. Robustness and Degradation of Agents under Noise

The deployment of agents in complex real-world environments has highlighted the critical importance of robustness (Li et al., 2025b; Levy et al., 2025; Agrawal et al., 2025; Fang et al., 2025). Recent studies indicate that agents suffer substantial performance degradation when faced with out-of-distribution scenarios or ambiguous user behaviors (Anghel et al., 2025; Wan et al., 2025; Herrera-Poyatos et al., 2025; Wang et al., 2025c). Various efforts have explored agent behavior under prompt perturbations or dialog variations (Li et al., 2025a; Deshpande et al., 2025; Qi et al., 2025). On the execution side, reliance on external tools introduces further instability, where agents usually encounter noisy, incomplete, or failed outputs (Xu et al., 2024b; Zhang et al., 2024b; Vuddanti et al., 2025), often leading to compounding errors (Yang et al., 2025; Zhu et al., 2025).

Nevertheless, existing evaluations (Nalbandyan et al., 2025; Wen et al., 2025; Yu et al., 2025) generally lack a systematic noise taxonomy and often focus on single noise sources. In addition, they typically only use the success rate as an evaluation metric. AgentNoiseBench introduces a unified framework covering both instruction and tool-execution noise and employs a trajectory-aware evaluation protocol to provide a multi-dimensional analysis to comprehensively characterize agent performance under noise.

## 5. Limitations

Despite the insights of our benchmark, several limitations remain. First, the evaluation targets language-centric, tool-augmented agents and may not generalize to agents with explicit planning, learned world models, or asynchronous tool environments. Second, while trajectory-level entropy is a useful diagnostic signal, it provides limited causal insight, as the analysis captures correlations between noise and instability. Third, trajectory-aware evaluation incurs computational overhead, and the benchmark lacks a scalable trade-off between diagnostic fidelity and cost, limiting its applicability in large-scale or real-world settings.

## 6. Future Work

These limitations motivate future research. First, we will study robustness-oriented training, such as reinforcement learning with noise-aware objectives, to improve agent stability under noisy interactions. Second, we plan to extend the benchmark to a wider range of agent architectures, task domains, and tool ecosystems. Finally, future work will develop analytical frameworks integrating trajectory-level diagnostics with causal and counterfactual analysis to better understand agent robustness.

## 7. Conclusion

In this work, we rethink the evaluation of LLM-based agents through the lens of real-world imperfections, introducing AgentNoiseBench to bridge the gap between controlled benchmarks and practical deployments. With our automated pipeline, we can explicitly incorporate real-world noise into

any agent-centric benchmarks while ensuring the tasks are still solvable. Our evaluation reveals that nearly all models suffer substantial performance degradation, with an average accuracy drop of 20.8%. We urge greater attention to real-world imperfections when training and evaluating agents.

## Impact Statement

This work introduces AgentNoiseBench, a systematic and standardized framework to evaluate agent robustness against realistic noise scenarios, lowering the entry barrier for developing more reliable and resilient autonomous systems. We critically challenge the common assumption that strong reasoning ability naturally translates into robustness, and instead demonstrate that agents remain especially susceptible to failures caused by tool-execution errors and intermediate-stage noise. Proactively diagnosing and addressing these specific vulnerabilities is essential for building agents that are not only capable, but also safe, trustworthy, and deployable in real-world settings.

## Acknowledgment

This research is supported by the New Generation Artificial Intelligence-National Science and Technology Major Project (2025ZD0123302), the National Research Foundation, Singapore, under its National Large Language Models Funding Initiative (AISG Award No: AISG-NMLP-2024-002). Any opinions, findings and conclusions or recommendations expressed in this material are those of the authors and do not reflect the views of National Research Foundation, Singapore.

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

## A. More Experimental Results

### A.1. Main Results

In this section, we conduct a more fine-grained analysis of the Agent Noise Benchmark results reported in Table 1. Specifically, we investigate how different types of user-side noise and tool-side noise impact mainstream agentic models across varied task scenarios.

Our analysis focuses on examining the granular effects of individual noise types on model performance. We systematically evaluate how each specific noise category—ranging from user ambiguity and topic shifts to tool errors and incomplete outputs—affects agent behavior across different application domains (airline, delivery, in-store, and retail). This fine-grained perspective allows us to identify which noise types pose the greatest challenges to current agentic systems and how these vulnerabilities vary across different task contexts and model architectures.

Detailed results are presented in Table 2, Table 3, Table 4, Table 5, Table 6, Table 7, Table 8, and Table 9. All experimental settings follow those described in Section 3.

Results show that all evaluated agents experience performance degradation under at least one noise type, with sensitivity patterns depending on both noise origin and model architecture. User-side noise primarily affects intent understanding and planning stages, leading to instruction misinterpretation or suboptimal plans. Tool-side noise more directly impairs execution reliability by introducing uncertainty in tool outputs. Notably, certain noise types such as conflict and incomplete information consistently cause more severe performance drops across models, while others like redundancy show more varied impacts depending on the specific agent architecture.

These findings emphasize the importance of systematically assessing and enhancing agent robustness in noisy environments at a granular level, and highlight the need for noise-type-specific mitigation strategies tailored to address the unique challenges posed by different categories of user-side and tool-side disturbances.

| Model | Airline | | | | | |
|---|---|---|---|---|---|---|
| | clean | ambiguity | topic_shift | conflict | redundancy | boundary |
| **Non-think Models** | | | | | | |
| DeepSeek-V3-0324 | $50.0_{\pm 2.3}$ | $45.8_{\pm 1.9}$ | $50.0_{\pm 2.1}$ | $50.0_{\pm 1.7}$ | $45.8_{\pm 2.4}$ | $50.0_{\pm 1.8}$ |
| Doubao-Seed-1.6 | $62.5_{\pm 2.2}$ | $54.2_{\pm 1.8}$ | $54.2_{\pm 2.0}$ | $54.2_{\pm 2.5}$ | $54.2_{\pm 2.3}$ | $62.5_{\pm 1.9}$ |
| GPT-4.1 | $60.4_{\pm 2.1}$ | $41.7_{\pm 2.6}$ | $47.9_{\pm 1.8}$ | $47.9_{\pm 2.2}$ | $41.7_{\pm 2.7}$ | $60.4_{\pm 2.0}$ |
| DeepSeek-V3.1 | $68.8_{\pm 1.9}$ | $54.2_{\pm 2.4}$ | $56.3_{\pm 2.2}$ | $52.1_{\pm 2.6}$ | $54.2_{\pm 1.8}$ | $68.8_{\pm 2.5}$ |
| gemini-2.5-pro | $77.1_{\pm 2.2}$ | $54.2_{\pm 2.1}$ | $60.4_{\pm 2.3}$ | $62.5_{\pm 1.9}$ | $62.5_{\pm 2.0}$ | $77.1_{\pm 2.4}$ |
| Qwen3-Max | $66.7_{\pm 2.4}$ | $41.7_{\pm 2.2}$ | $54.2_{\pm 2.5}$ | $54.2_{\pm 1.8}$ | $54.2_{\pm 2.1}$ | $66.7_{\pm 2.3}$ |
| LongCat-Flash-Chat | $66.7_{\pm 2.0}$ | $50.0_{\pm 2.3}$ | $58.3_{\pm 2.2}$ | $56.3_{\pm 1.9}$ | $56.3_{\pm 2.4}$ | $66.7_{\pm 2.1}$ |
| Claude-4-Sonnet | $77.1_{\pm 2.5}$ | $66.7_{\pm 2.0}$ | $60.4_{\pm 2.4}$ | $58.3_{\pm 2.1}$ | $66.7_{\pm 2.3}$ | $77.1_{\pm 1.9}$ |
| Claude-4.5-Sonnet | $75.0_{\pm 2.1}$ | $68.8_{\pm 2.5}$ | $60.4_{\pm 1.9}$ | $58.3_{\pm 2.2}$ | $60.4_{\pm 2.6}$ | $75.0_{\pm 2.3}$ |
| **Thinking Models** | | | | | | |
| Qwen3-max (w/ thinking) | $89.6_{\pm 2.4}$ | $70.8_{\pm 2.2}$ | $58.3_{\pm 2.6}$ | $58.3_{\pm 2.0}$ | $52.1_{\pm 2.3}$ | $89.6_{\pm 2.5}$ |
| Gemini-2.5-Flash (think on) | $77.1_{\pm 2.3}$ | $56.3_{\pm 1.9}$ | $54.2_{\pm 2.2}$ | $54.2_{\pm 2.4}$ | $56.3_{\pm 2.1}$ | $77.1_{\pm 2.6}$ |
| Doubao-Seed-1.6-Thinking | $77.1_{\pm 2.5}$ | $54.2_{\pm 2.1}$ | $60.4_{\pm 2.3}$ | $60.4_{\pm 1.9}$ | $56.3_{\pm 2.4}$ | $77.1_{\pm 2.0}$ |
| GLM-4.6 (w/ thinking) | $75.0_{\pm 2.3}$ | $77.1_{\pm 2.4}$ | $66.7_{\pm 2.0}$ | $66.7_{\pm 2.2}$ | $68.8_{\pm 2.5}$ | $75.0_{\pm 2.1}$ |
| deepseek-R1-0528 (w/ thinking) | $70.8_{\pm 2.4}$ | $52.1_{\pm 2.6}$ | $52.1_{\pm 2.2}$ | $52.1_{\pm 2.5}$ | $41.7_{\pm 2.0}$ | $70.8_{\pm 2.3}$ |
| gpt-5.2 (w/ thinking) | $79.2_{\pm 2.5}$ | $60.4_{\pm 2.3}$ | $68.8_{\pm 2.1}$ | $66.7_{\pm 2.4}$ | $66.7_{\pm 2.6}$ | $79.2_{\pm 2.2}$ |
| o3 (w/ thinking) | $77.1_{\pm 2.2}$ | $72.9_{\pm 2.4}$ | $66.7_{\pm 2.3}$ | $66.7_{\pm 2.1}$ | $64.6_{\pm 2.5}$ | $77.1_{\pm 2.6}$ |

*Table 2.* Model performance on airline scene across different user noise types (mean ± std).

| Model | Delivery | | | | | |
|---|---|---|---|---|---|---|
| | clean | ambiguity | topic_shift | conflict | redundancy | boundary |
| **Non-think Models** | | | | | | |
| DeepSeek-V3-0324 | 42.3±2.6 | 33.0±1.9 | 38.0±2.4 | 34.0±1.7 | 38.0±2.5 | 49.0±2.1 |
| Doubao-Seed-1.6 | 35.0±2.4 | 27.0±1.8 | 33.0±2.2 | 32.0±2.5 | 37.0±2.3 | 38.0±1.9 |
| GPT-4.1 | 38.0±2.1 | 33.0±2.6 | 40.0±1.8 | 35.0±2.4 | 38.0±1.9 | 38.0±2.5 |
| DeepSeek-V3.1 | 33.0±2.5 | 22.0±2.0 | 32.0±2.4 | 38.0±1.8 | 39.0±2.6 | 42.6±2.3 |
| gemini-2.5-pro | 43.0±2.3 | 44.0±2.1 | 46.0±2.5 | 48.0±2.2 | 48.0±1.9 | 50.0±2.4 |
| Qwen3-Max | 27.0±2.6 | 27.0±2.1 | 25.0±2.4 | 24.0±2.3 | 31.0±2.0 | 31.0±1.8 |
| LongCat-Flash-Chat | 35.0±2.3 | 42.0±1.9 | 44.0±2.1 | 41.0±2.5 | 42.0±2.4 | 44.0±2.0 |
| Claude-4-Sonnet | 35.0±2.4 | 29.0±2.2 | 42.0±2.6 | 39.0±1.8 | 40.0±2.5 | 45.0±2.1 |
| Claude-4.5-Sonnet | 47.0±2.0 | 48.0±2.5 | 53.0±2.3 | 47.0±2.6 | 54.0±2.1 | 53.6±1.9 |
| **Thinking Models** | | | | | | |
| DeepSeek-R1-0528 (w/ thinking) | 41.0±2.6 | 35.0±2.4 | 48.0±2.5 | 41.0±2.0 | 50.0±2.3 | 44.0±2.2 |
| Doubao-Seed-1.6-Thinking | 35.0±2.5 | 35.0±2.0 | 41.0±2.4 | 32.0±2.6 | 37.0±2.3 | 37.0±2.1 |
| Gemini-2.5-Flash (think on) | 26.0±2.4 | 18.0±2.1 | 35.0±2.3 | 29.0±2.5 | 21.0±2.6 | 34.0±1.9 |
| gemini-2.5-pro (w/ thinking) | 43.0±2.3 | 44.0±2.7 | 46.0±2.4 | 48.0±2.1 | 48.0±2.5 | 50.0±2.2 |
| GLM-4.5 (w/ thinking) | 37.0±2.4 | 38.0±2.6 | 45.0±2.2 | 43.0±2.3 | 39.6±2.1 | 46.0±2.5 |
| GLM-4.6 (w/ thinking) | 38.6±2.3 | 29.0±2.6 | 47.5±2.4 | 45.2±2.2 | 41.4±2.5 | 47.0±2.1 |
| GPT-5.2 (w/ thinking) | 51.0±2.5 | 53.0±2.3 | 53.0±2.6 | 48.0±2.4 | 51.0±2.2 | 54.1±2.7 |
| o3 (w/ thinking) | 39.0±2.3 | 39.4±2.6 | 55.0±2.4 | 47.0±2.2 | 56.0±2.5 | 46.0±2.1 |

*Table 3.* Model performance on delivery scene across different user noise types (mean ± std).

| Model | Instore | | | | | |
|---|---|---|---|---|---|---|
| | clean | ambiguity | topic_shift | conflict | redundancy | boundary |
| **Non-think Models** | | | | | | |
| DeepSeek-V3-0324 | 30.0±2.5 | 18.6±1.8 | 25.0±2.3 | 16.0±1.9 | 22.0±2.6 | 24.0±2.1 |
| Doubao-Seed-1.6 | 53.9±2.4 | 38.2±1.7 | 48.2±2.2 | 35.6±2.5 | 42.3±2.3 | 50.2±1.8 |
| GPT-4.1 | 59.0±2.1 | 42.0±2.6 | 52.0±1.8 | 40.0±2.4 | 48.0±1.9 | 54.0±2.5 |
| DeepSeek-V3.1 | 55.0±2.6 | 39.6±2.0 | 49.4±2.4 | 37.6±1.7 | 45.5±2.5 | 51.6±2.3 |
| gemini-2.5-pro | 53.0±2.2 | 37.6±2.3 | 47.1±2.5 | 38.0±2.1 | 44.0±1.9 | 49.0±2.4 |
| Qwen3-Max | 57.8±2.6 | 40.0±2.1 | 50.8±2.4 | 38.0±2.3 | 46.0±2.0 | 52.6±1.8 |
| LongCat-Flash-Chat | 62.0±2.3 | 43.2±1.9 | 54.0±2.1 | 42.0±2.5 | 50.0±2.4 | 57.0±2.0 |
| Claude-4-Sonnet | 56.0±2.4 | 39.0±2.2 | 50.0±2.6 | 38.0±1.8 | 46.0±2.5 | 52.0±2.1 |
| Claude-4.5-Sonnet | 58.0±2.0 | 41.6±2.5 | 52.0±2.3 | 40.0±2.6 | 48.0±2.1 | 54.0±1.9 |
| **Thinking Models** | | | | | | |
| DeepSeek-R1-0528 (w/ thinking) | 59.0±2.6 | 42.7±2.4 | 53.0±2.5 | 41.0±2.0 | 49.0±2.3 | 55.0±2.2 |
| Doubao-Seed-1.6-Thinking | 45.0±2.5 | 32.5±2.0 | 40.0±2.4 | 30.0±2.6 | 37.0±2.3 | 42.0±2.1 |
| Gemini-2.5-Flash (think on) | 26.0±2.4 | 16.0±2.1 | 20.0±2.3 | 14.0±2.5 | 18.0±2.6 | 22.0±1.9 |
| gemini-2.5-pro (w/ thinking) | 57.0±2.3 | 40.6±2.7 | 51.0±2.4 | 39.0±2.1 | 47.0±2.5 | 53.0±2.2 |
| GLM-4.5 (w/ thinking) | 64.0±2.4 | 46.0±2.6 | 57.0±2.2 | 44.0±2.3 | 52.0±2.1 | 59.0±2.5 |
| GLM-4.6 (w/ thinking) | 65.8±2.3 | 48.5±2.6 | 59.4±2.4 | 46.0±2.2 | 54.0±2.5 | 61.0±2.1 |
| GPT-5.2 (w/ thinking) | 65.0±2.5 | 47.9±2.3 | 58.0±2.6 | 45.0±2.4 | 53.0±2.2 | 60.0±2.7 |
| o3 (w/ thinking) | 64.0±2.3 | 47.5±2.6 | 58.0±2.4 | 46.0±2.2 | 54.0±2.5 | 60.0±2.1 |

*Table 4.* Model performance on instore scene across different user noise types (mean ± std).

| Model | Retail | | | | | |
|---|---|---|---|---|---|---|
| | **clean** | **ambiguity** | **topic_shift** | **conflict** | **redundancy** | **boundary** |
| **Non-think Models** | | | | | | |
| DeepSeek-V3-0324 | 66.1±2.4 | 47.8±1.8 | 55.3±2.3 | 51.7±1.9 | 57.6±2.5 | 59.4±2.1 |
| Doubao-Seed-1.6 | 70.5±2.3 | 51.8±1.7 | 59.9±2.2 | 56.2±2.4 | 62.1±2.1 | 63.8±1.8 |
| GPT-4.1 | 74.1±2.1 | 56.2±2.6 | 64.1±1.8 | 60.0±2.5 | 65.6±1.9 | 68.2±2.3 |
| DeepSeek-V3.1 | 72.3±2.5 | 53.7±2.0 | 62.2±2.4 | 58.1±1.7 | 64.1±2.6 | 66.1±2.3 |
| gemini-2.5-pro | 66.1±2.2 | 48.5±2.3 | 56.9±2.5 | 52.3±2.1 | 58.4±1.9 | 60.4±2.4 |
| Qwen3-Max | 71.4±2.6 | 52.7±2.1 | 61.0±2.4 | 56.4±2.3 | 62.6±2.0 | 65.0±1.8 |
| LongCat-Flash-Chat | 79.5±2.3 | 59.6±1.9 | 68.2±2.1 | 63.4±2.5 | 69.5±2.4 | 71.8±2.0 |
| Claude-4-Sonnet | 73.2±2.4 | 54.6±2.2 | 62.8±2.6 | 59.2±1.8 | 64.3±2.5 | 66.9±2.1 |
| Claude-4.5-Sonnet | 79.5±2.0 | 59.8±2.5 | 68.6±2.3 | 63.6±2.6 | 69.4±2.1 | 72.2±1.9 |
| **Thinking Models** | | | | | | |
| DeepSeek-R1-0528 (w/ thinking) | 57.1±2.6 | 41.7±2.4 | 49.3±2.5 | 45.9±2.0 | 51.9±2.3 | 53.4±2.2 |
| Doubao-Seed-1.6-Thinking | 58.0±2.5 | 42.4±2.0 | 50.1±2.4 | 46.8±2.6 | 53.1±2.3 | 55.0±2.1 |
| Gemini-2.5-Flash (think on) | 69.6±2.4 | 51.1±2.1 | 59.3±2.3 | 55.3±2.5 | 61.6±2.6 | 63.6±1.9 |
| gemini-2.5-pro (w/ thinking) | 73.2±2.3 | 55.2±2.7 | 63.1±2.4 | 59.1±2.1 | 64.8±2.5 | 66.9±2.2 |
| GLM-4.5 (w/ thinking) | 80.4±2.4 | 62.1±2.6 | 70.3±2.2 | 65.8±2.3 | 71.5±2.1 | 74.0±2.5 |
| GLM-4.6 (w/ thinking) | 84.8±2.3 | 65.3±2.6 | 73.8±2.4 | 68.3±2.2 | 74.3±2.5 | 76.5±2.1 |
| GPT-5.2 (w/ thinking) | 73.2±2.5 | 55.2±2.3 | 63.1±2.6 | 59.1±2.4 | 64.8±2.2 | 66.8±2.7 |
| o3 (w/ thinking) | 80.0±2.3 | 62.2±2.6 | 70.9±2.4 | 67.0±2.2 | 72.9±2.5 | 75.1±2.1 |

*Table 5.* Model performance on retail scene across different user noise types (mean ± std).

| Model | Airline | | | | | |
|---|---|---|---|---|---|---|
| | **clean** | **error** | **failure** | **incomplete** | **induce** | **redundancy** |
| **Non-think Models** | | | | | | |
| DeepSeek-V3-0324 | 50.0±2.0 | 48.5±2.2 | 50.0±1.6 | 50.0±2.1 | 45.8±2.0 | 45.8±2.3 |
| Doubao-Seed-1.6 | 62.5±1.9 | 60.9±2.1 | 54.2±1.5 | 47.9±2.0 | 54.2±1.8 | 54.2±2.1 |
| GPT-4.1 | 60.4±2.4 | 61.2±2.5 | 47.9±2.0 | 43.8±2.3 | 41.8±2.2 | 47.9±2.4 |
| DeepSeek-V3.1 | 68.8±2.1 | 68.3±2.3 | 52.1±1.8 | 56.3±2.2 | 54.2±2.1 | 54.2±2.2 |
| gemini-2.5-pro | 77.1±2.5 | 75.9±2.4 | 60.4±1.9 | 52.1±2.3 | 54.2±2.4 | 62.5±2.5 |
| Qwen3-Max | 66.7±2.3 | 65.1±2.4 | 54.2±1.9 | 45.8±2.2 | 41.7±2.3 | 54.2±2.4 |
| LongCat-Flash-Chat | 66.7±2.1 | 67.6±2.2 | 58.3±1.7 | 54.2±2.1 | 50.0±2.0 | 56.3±2.2 |
| Claude-4-Sonnet | 77.1±2.2 | 74.7±2.3 | 60.4±1.9 | 58.3±2.2 | 66.7±2.1 | 66.7±2.3 |
| Claude-4.5-Sonnet | 75.0±2.4 | 78.1±2.4 | 58.3±2.0 | 60.4±2.3 | 68.8±2.3 | 60.4±2.5 |
| **Thinking Models** | | | | | | |
| Qwen3-max (w/ thinking) | 89.6±2.6 | 89.1±2.5 | 58.3±2.1 | 60.4±2.5 | 70.8±2.6 | 52.1±2.7 |
| Gemini-2.5-Flash (think on) | 77.1±2.0 | 76.6±2.2 | 54.2±1.7 | 47.9±2.1 | 56.3±2.0 | 56.3±2.2 |
| Doubao-Seed-1.6-Thinking | 77.1±2.1 | 76.6±2.3 | 60.4±1.8 | 60.4±2.2 | 54.2±2.1 | 56.3±2.3 |
| GLM-4.6 (w/ thinking) | 77.1±2.6 | 74.5±2.5 | 66.7±2.1 | 62.5±2.5 | 77.1±2.6 | 68.8±2.7 |
| deepseek-R1-0528 (w/ thinking) | 70.8±2.2 | 70.3±2.3 | 52.1±1.9 | 50.0±2.3 | 52.1±2.2 | 41.7±2.4 |
| gpt-5.2 (w/ thinking) | 79.2±2.8 | 78.7±2.7 | 66.7±2.3 | 68.8±2.7 | 60.4±2.8 | 66.7±2.9 |
| o3 (w/ thinking) | 77.1±2.5 | 76.6±2.5 | 66.7±2.1 | 54.2±2.5 | 72.9±2.6 | 64.6±2.7 |

*Table 6.* Model performance on airline scene across different tool noise types (mean ± std).

| Model | Delivery | | | | | |
|---|---|---|---|---|---|---|
| | clean | error | failure | incomplete | induce | redundancy |
| **Non-think Models** | | | | | | |
| DeepSeek-V3-0324 | 42.3±2.4 | 49.0±1.8 | 16.8±2.2 | 37.2±1.9 | 39.8±2.6 | 40.5±2.1 |
| Doubao-Seed-1.6 | 39.8±2.3 | 38.3±1.7 | 14.2±2.0 | 34.6±2.4 | 36.3±2.2 | 37.9±1.8 |
| GPT-4.1 | 48.6±2.1 | 38.0±2.6 | 22.1±1.8 | 42.9±2.5 | 45.7±1.9 | 46.3±2.3 |
| DeepSeek-V3.1 | 44.2±2.5 | 42.6±2.0 | 18.5±2.3 | 39.4±1.7 | 41.6±2.4 | 42.8±2.6 |
| gemini-2.5-pro | 51.3±2.2 | 50.0±2.3 | 21.3±2.5 | 46.7±2.1 | 49.2±1.9 | 50.1±2.4 |
| Qwen3-Max | 47.5±2.6 | 31.0±2.1 | 19.8±2.4 | 41.3±2.3 | 44.6±2.0 | 45.2±1.8 |
| LongCat-Flash-Chat | 40.7±2.3 | 44.0±1.9 | 15.9±2.1 | 35.8±2.5 | 38.2±2.4 | 39.4±2.0 |
| Claude-4-Sonnet | 45.9±2.4 | 45.0±2.2 | 20.3±2.6 | 40.8±1.8 | 43.1±2.5 | 44.6±2.1 |
| Claude-4.5-Sonnet | 49.3±2.0 | 53.6±2.5 | 22.5±2.3 | 44.2±2.6 | 46.8±2.1 | 47.9±1.9 |
| **Thinking Models** | | | | | | |
| Qwen3-max (w/ thinking) | 58.7±2.3 | 44.3±2.7 | 25.6±2.4 | 52.8±2.1 | 55.9±2.5 | 56.3±2.2 |
| Gemini-2.5-Flash (think on) | 41.2±2.4 | 29.8±2.1 | 16.4±2.3 | 36.7±2.5 | 38.9±2.6 | 39.7±1.9 |
| Doubao-Seed-1.6-Thinking | 43.9±2.5 | 32.7±2.0 | 18.9±2.4 | 39.4±2.6 | 41.3±2.3 | 42.1±2.1 |
| GLM-4.6 (w/ thinking) | 54.8±2.4 | 42.3±2.6 | 24.9±2.2 | 50.1±2.3 | 52.7±2.1 | 53.2±2.5 |
| deepseek-R1-0528 (w/ thinking) | 46.3±2.6 | 35.9±2.4 | 21.4±2.5 | 42.1±2.0 | 44.7±2.3 | 45.3±2.2 |
| gpt-5.2 (w/ thinking) | 62.1±2.5 | 48.6±2.3 | 28.7±2.6 | 56.3±2.4 | 59.4±2.2 | 60.2±2.7 |
| o3 (w/ thinking) | 53.7±2.3 | 41.5±2.6 | 24.2±2.4 | 49.3±2.2 | 51.8±2.5 | 52.5±2.1 |

*Table 7.* Model performance on delivery scene across different tool noise types (mean ± std).

| Model | Instore | | | | | |
|---|---|---|---|---|---|---|
| | clean | error | failure | incomplete | induce | redundancy |
| **Non-think Models** | | | | | | |
| DeepSeek-V3-0324 | 30.0±2.4 | 29.0±1.8 | 24.0±2.3 | 8.0±1.9 | 28.0±2.5 | 30.0±2.1 |
| Doubao-Seed-1.6 | 53.9±2.3 | 46.5±1.7 | 33.0±2.2 | 35.7±2.4 | 45.0±2.1 | 53.9±1.8 |
| GPT-4.1 | 59.0±2.1 | 56.0±2.6 | 41.0±1.8 | 13.0±2.5 | 48.0±1.9 | 59.0±2.3 |
| DeepSeek-V3.1 | 55.0±2.5 | 59.0±2.0 | 36.0±2.4 | 30.0±1.7 | 49.0±2.6 | 55.0±2.3 |
| gemini-2.5-pro | 57.0±2.2 | 55.0±2.3 | 34.0±2.5 | 24.0±2.1 | 46.0±1.9 | 57.0±2.4 |
| Qwen3-Max | 57.8±2.6 | 50.0±2.1 | 27.0±2.4 | 11.0±2.3 | 44.0±2.0 | 57.8±1.8 |
| LongCat-Flash-Chat | 62.0±2.3 | 63.0±1.9 | 38.0±2.1 | 29.0±2.5 | 49.0±2.4 | 62.0±2.0 |
| Claude-4-Sonnet | 58.3±2.4 | 53.9±2.2 | 45.5±2.6 | 25.2±1.8 | 46.4±2.5 | 58.3±2.1 |
| Claude-4.5-Sonnet | 61.3±2.0 | 55.8±2.5 | 44.0±2.3 | 41.6±2.6 | 55.0±2.1 | 61.3±1.9 |
| **Thinking Models** | | | | | | |
| Qwen3-max (w/ thinking) | 57.0±2.3 | 55.0±2.7 | 34.0±2.4 | 24.0±2.1 | 46.0±2.5 | 57.0±2.2 |
| DeepSeek-R1-0528 (w/ thinking) | 59.0±2.6 | 56.0±2.4 | 46.0±2.5 | 25.0±2.0 | 46.0±2.3 | 59.0±2.2 |
| Doubao-Seed-1.6-Thinking | 45.0±2.5 | 45.0±2.0 | 36.0±2.4 | 36.0±2.6 | 36.0±2.3 | 45.0±2.1 |
| Gemini-2.5-Flash (think on) | 26.0±2.4 | 24.0±2.1 | 10.0±2.3 | 2.0±2.5 | 18.0±2.6 | 26.0±1.9 |
| GLM-4.6 (w/ thinking) | 65.8±2.4 | 59.0±2.6 | 57.0±2.2 | 53.5±2.3 | 45.0±2.1 | 65.8±2.5 |
| GPT-5.2 (w/ thinking) | 65.0±2.5 | 57.0±2.3 | 54.0±2.6 | 42.0±2.4 | 49.0±2.2 | 65.0±2.7 |
| o3 (w/ thinking) | 64.0±2.3 | 59.0±2.6 | 53.0±2.4 | 62.0±2.2 | 58.0±2.5 | 64.0±2.1 |

*Table 8.* Model performance on instore scene across different tool noise types (mean ± std).

| Model | Retail | | | | | |
|---|---|---|---|---|---|---|
| | clean | error | failure | incomplete | induce | redundancy |
| Non-think Models | | | | | | |
| DeepSeek-V3-0324 | $66.1_{\pm2.4}$ | $38.5_{\pm1.8}$ | $24.1_{\pm2.3}$ | $29.5_{\pm1.9}$ | $42.3_{\pm2.5}$ | $58.2_{\pm2.1}$ |
| Doubao-Seed-1.6 | $70.5_{\pm2.3}$ | $45.2_{\pm1.7}$ | $28.6_{\pm2.2}$ | $35.6_{\pm2.4}$ | $48.9_{\pm2.1}$ | $62.4_{\pm1.8}$ |
| GPT-4.1 | $74.1_{\pm2.1}$ | $52.9_{\pm2.6}$ | $26.8_{\pm1.8}$ | $42.7_{\pm2.5}$ | $55.3_{\pm1.9}$ | $66.8_{\pm2.3}$ |
| DeepSeek-V3.1 | $72.3_{\pm2.5}$ | $48.4_{\pm2.0}$ | $33.9_{\pm2.4}$ | $38.8_{\pm1.7}$ | $51.8_{\pm2.6}$ | $64.5_{\pm2.3}$ |
| gemini-2.5-pro | $66.1_{\pm2.2}$ | $41.2_{\pm2.3}$ | $28.6_{\pm2.5}$ | $34.0_{\pm2.1}$ | $47.1_{\pm1.9}$ | $59.3_{\pm2.4}$ |
| Qwen3-Max | $71.4_{\pm2.6}$ | $46.4_{\pm2.1}$ | $27.7_{\pm2.4}$ | $37.1_{\pm2.3}$ | $49.3_{\pm2.0}$ | $63.7_{\pm1.8}$ |
| LongCat-Flash-Chat | $79.5_{\pm2.3}$ | $54.3_{\pm1.9}$ | $33.9_{\pm2.1}$ | $43.1_{\pm2.5}$ | $57.2_{\pm2.4}$ | $71.8_{\pm2.0}$ |
| Claude-4-Sonnet | $73.2_{\pm2.4}$ | $49.3_{\pm2.2}$ | $29.5_{\pm2.6}$ | $39.9_{\pm1.8}$ | $52.0_{\pm2.5}$ | $65.6_{\pm2.1}$ |
| Claude-4.5-Sonnet | $79.5_{\pm2.0}$ | $55.5_{\pm2.5}$ | $39.3_{\pm2.3}$ | $45.3_{\pm2.6}$ | $58.1_{\pm2.1}$ | $72.4_{\pm1.9}$ |
| Thinking Models | | | | | | |
| Qwen3-Max (w/ thinking) | $75.9_{\pm2.3}$ | $51.1_{\pm2.7}$ | $40.2_{\pm2.4}$ | $42.5_{\pm2.1}$ | $56.5_{\pm2.5}$ | $68.9_{\pm2.2}$ |
| Gemini-2.5-Flash (think on) | $69.6_{\pm2.4}$ | $44.8_{\pm2.1}$ | $25.0_{\pm2.3}$ | $36.0_{\pm2.5}$ | $50.3_{\pm2.6}$ | $62.1_{\pm1.9}$ |
| Doubao-Seed-1.6-Thinking | $58.0_{\pm2.5}$ | $35.1_{\pm2.0}$ | $20.5_{\pm2.4}$ | $28.8_{\pm2.6}$ | $41.7_{\pm2.3}$ | $52.4_{\pm2.1}$ |
| GLM-4.6 (w/ thinking) | $84.8_{\pm2.4}$ | $60.0_{\pm2.6}$ | $42.0_{\pm2.2}$ | $48.5_{\pm2.3}$ | $63.2_{\pm2.1}$ | $77.3_{\pm2.5}$ |
| DeepSeek-R1-0528 (w/ thinking) | $57.1_{\pm2.6}$ | $34.4_{\pm2.4}$ | $25.0_{\pm2.5}$ | $31.6_{\pm2.0}$ | $42.6_{\pm2.3}$ | $51.8_{\pm2.2}$ |
| GPT-5.2 (w/ thinking) | $73.2_{\pm2.5}$ | $49.9_{\pm2.3}$ | $34.8_{\pm2.6}$ | $40.8_{\pm2.4}$ | $54.5_{\pm2.2}$ | $66.7_{\pm2.7}$ |
| o3 (w/ thinking) | $80.0_{\pm2.3}$ | $56.0_{\pm2.6}$ | $48.0_{\pm2.4}$ | $47.0_{\pm2.2}$ | $60.0_{\pm2.5}$ | $73.5_{\pm2.1}$ |

*Table 9.* Model performance on retail scene across different tool noise types (mean ± std).

| Model | HotpotQA | | | | | |
|---|---|---|---|---|---|---|
| | clean | error | failure | incomplete | induce | redundancy |
| Non-think models | | | | | | |
| DeepSeek-V3-0324 | $0.21_{\pm0.08}$ | $0.19_{\pm0.02}$ | $0.16_{\pm0.06}$ | $0.23_{\pm0.04}$ | $0.20_{\pm0.06}$ | $0.21_{\pm0.02}$ |
| Gemini-2.5-Flash (think off) | $0.29_{\pm0.04}$ | $0.28_{\pm0.06}$ | $0.12_{\pm0.02}$ | $0.25_{\pm0.02}$ | $0.23_{\pm0.02}$ | $0.24_{\pm0.06}$ |
| GPT-4.1 | $0.33_{\pm0.04}$ | $0.29_{\pm0.01}$ | $0.30_{\pm0.06}$ | $0.29_{\pm0.04}$ | $0.29_{\pm0.04}$ | $0.29_{\pm0.02}$ |
| DeepSeek-V3.1 | $0.40_{\pm0.06}$ | $0.31_{\pm0.06}$ | $0.29_{\pm0.02}$ | $0.38_{\pm0.04}$ | $0.37_{\pm0.04}$ | $0.38_{\pm0.04}$ |
| DeepSeek-V3.2-Exp (w/o thinking) | $0.43_{\pm0.02}$ | $0.37_{\pm0.02}$ | $0.32_{\pm0.06}$ | $0.42_{\pm0.04}$ | $0.41_{\pm0.02}$ | $0.43_{\pm0.06}$ |
| gpt-3.5-turbo (w/o thinking) | $0.18_{\pm0.04}$ | $0.14_{\pm0.03}$ | $0.13_{\pm0.01}$ | $0.16_{\pm0.02}$ | $0.17_{\pm0.06}$ | $0.13_{\pm0.04}$ |
| GLM-4.6 | $0.41_{\pm0.06}$ | $0.37_{\pm0.02}$ | $0.05_{\pm0.02}$ | $0.40_{\pm0.04}$ | $0.43_{\pm0.01}$ | $0.41_{\pm0.08}$ |
| LongCat-Flash-Chat | $0.36_{\pm0.04}$ | $0.26_{\pm0.02}$ | $0.22_{\pm0.01}$ | $0.34_{\pm0.02}$ | $0.32_{\pm0.01}$ | $0.32_{\pm0.06}$ |
| Claude-4-Sonnet | $0.34_{\pm0.04}$ | $0.28_{\pm0.08}$ | $0.24_{\pm0.04}$ | $0.34_{\pm0.04}$ | $0.35_{\pm0.02}$ | $0.33_{\pm0.01}$ |
| Claude-4.5-Sonnet | $0.35_{\pm0.03}$ | $0.32_{\pm0.02}$ | $0.23_{\pm0.06}$ | $0.33_{\pm0.04}$ | $0.36_{\pm0.06}$ | $0.35_{\pm0.06}$ |
| Thinking Models | | | | | | |
| Qwen3-max (w/ thinking) | $0.43_{\pm0.01}$ | $0.33_{\pm0.02}$ | $0.14_{\pm0.06}$ | $0.41_{\pm0.01}$ | $0.36_{\pm0.04}$ | $0.43_{\pm0.06}$ |
| Gemini-2.5-Flash (think on) | $0.27_{\pm0.04}$ | $0.21_{\pm0.04}$ | $0.13_{\pm0.02}$ | $0.25_{\pm0.06}$ | $0.23_{\pm0.08}$ | $0.26_{\pm0.08}$ |
| GLM-4.5 (w/ thinking) | $0.44_{\pm0.02}$ | $0.35_{\pm0.04}$ | $0.27_{\pm0.04}$ | $0.41_{\pm0.01}$ | $0.44_{\pm0.02}$ | $0.37_{\pm0.05}$ |
| GLM-4.6 (w/ thinking) | $0.45_{\pm0.04}$ | $0.34_{\pm0.02}$ | $0.26_{\pm0.03}$ | $0.39_{\pm0.01}$ | $0.39_{\pm0.08}$ | $0.40_{\pm0.06}$ |

*Table 10.* Model Performance on HotpotQA across Different Noise Types (mean ± std).

| Model | Wiki | | | | | |
|---|---|---|---|---|---|---|
| | clean | error | failure | incomplete | induce | redundancy |
| **Non-think models** | | | | | | |
| DeepSeek-V3-0324 | $0.22_{\pm0.02}$ | $0.18_{\pm0.04}$ | $0.06_{\pm0.04}$ | $0.20_{\pm0.02}$ | $0.20_{\pm0.06}$ | $0.19_{\pm0.02}$ |
| Gemini-2.5-Flash (think off) | $0.27_{\pm0.04}$ | $0.20_{\pm0.06}$ | $0.03_{\pm0.08}$ | $0.24_{\pm0.04}$ | $0.23_{\pm0.06}$ | $0.25_{\pm0.06}$ |
| GPT-4.1 | $0.25_{\pm0.02}$ | $0.22_{\pm0.01}$ | $0.29_{\pm0.08}$ | $0.22_{\pm0.02}$ | $0.16_{\pm0.02}$ | $0.24_{\pm0.04}$ |
| DeepSeek-V3.1 | $0.44_{\pm0.02}$ | $0.42_{\pm0.03}$ | $0.30_{\pm0.02}$ | $0.44_{\pm0.04}$ | $0.47_{\pm0.06}$ | $0.47_{\pm0.04}$ |
| DeepSeek-V3.2-Exp (w/o thinking) | $0.44_{\pm0.02}$ | $0.42_{\pm0.03}$ | $0.30_{\pm0.02}$ | $0.44_{\pm0.04}$ | $0.47_{\pm0.06}$ | $0.47_{\pm0.04}$ |
| gpt-3.5-turbo (w/o thinking) | $0.44_{\pm0.02}$ | $0.42_{\pm0.03}$ | $0.30_{\pm0.02}$ | $0.44_{\pm0.04}$ | $0.47_{\pm0.06}$ | $0.47_{\pm0.04}$ |
| GLM-4.6 | $0.51_{\pm0.06}$ | $0.44_{\pm0.04}$ | $0.28_{\pm0.06}$ | $0.49_{\pm0.02}$ | $0.46_{\pm0.06}$ | $0.50_{\pm0.04}$ |
| LongCat-Flash-Chat | $0.40_{\pm0.01}$ | $0.32_{\pm0.02}$ | $0.16_{\pm0.07}$ | $0.37_{\pm0.02}$ | $0.39_{\pm0.04}$ | $0.39_{\pm0.02}$ |
| Claude-4-Sonnet | $0.40_{\pm0.02}$ | $0.32_{\pm0.03}$ | $0.10_{\pm0.04}$ | $0.38_{\pm0.03}$ | $0.38_{\pm0.02}$ | $0.31_{\pm0.03}$ |
| Claude-4.5-Sonnet | $0.32_{\pm0.04}$ | $0.29_{\pm0.02}$ | $0.09_{\pm0.06}$ | $0.31_{\pm0.02}$ | $0.34_{\pm0.02}$ | $0.31_{\pm0.04}$ |
| **Thinking Models** | | | | | | |
| Qwen3-max (w/ thinking) | $0.56_{\pm0.02}$ | $0.36_{\pm0.02}$ | $0.18_{\pm0.07}$ | $0.44_{\pm0.04}$ | $0.51_{\pm0.04}$ | $0.40_{\pm0.09}$ |
| Gemini-2.5-Flash (think on) | $0.27_{\pm0.06}$ | $0.18_{\pm0.04}$ | $0.08_{\pm0.04}$ | $0.21_{\pm0.06}$ | $0.19_{\pm0.06}$ | $0.28_{\pm0.04}$ |
| GLM-4.5 (w/ thinking) | $0.46_{\pm0.02}$ | $0.35_{\pm0.08}$ | $0.19_{\pm0.07}$ | $0.43_{\pm0.03}$ | $0.44_{\pm0.04}$ | $0.45_{\pm0.08}$ |
| GLM-4.6 (w/ thinking) | $0.47_{\pm0.08}$ | $0.33_{\pm0.02}$ | $0.20_{\pm0.02}$ | $0.39_{\pm0.01}$ | $0.43_{\pm0.02}$ | $0.41_{\pm0.02}$ |

*Table 11.* Model Performance on HotpotQA across Different Noise Types (mean ± std).

## A.2. Impact of Noise Injection Stage

We also evaluate the performance degradation of the remaining four model families under noise injection at different stages:

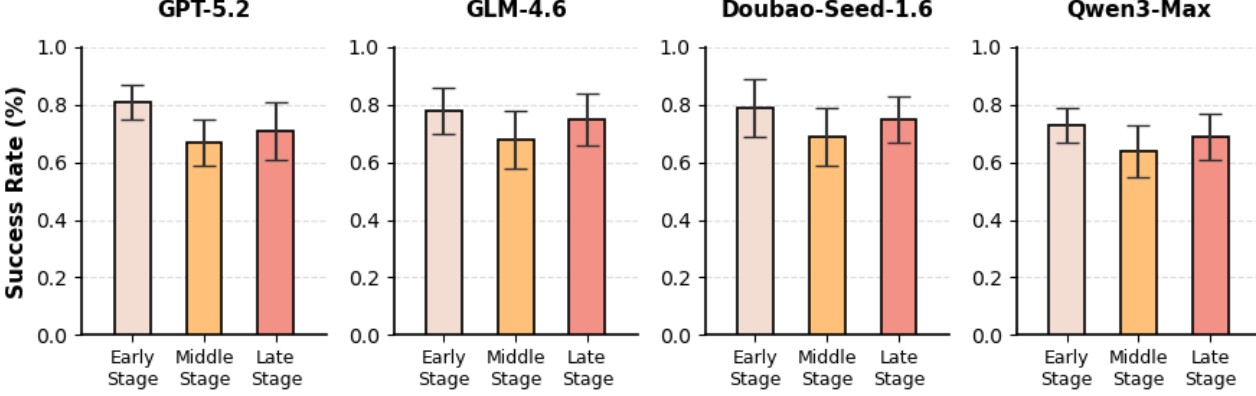

*Figure 6.* Impact of noise injection stages on model performance across different architectures.

# B. Noise Case Study

To facilitate a clearer understanding of our experimental pipeline, we randomly sample evaluation instances from different agent tasks and environments, and construct noisy variants using the noise injection mechanisms defined in the Agent Noise Benchmark. These noisy examples are generated by perturbing either intermediate tool outputs, contextual information, or decision-relevant signals, depending on the specific noise configuration under study. For illustration purposes, we present representative examples in Example B (clean trajectory), Example B (noisy trajectory with mid-stage noise injection), and additional cases in subsequent examples. The following are examples of user-side noise:

---

**Clean User Instruction**

Hi, I have an upcoming flight and I need to know exactly how many suitcases I'm allowed to bring with my reservation. I have a lot to pack, so I really need the correct number. Can you help me with that?

---

**Ambiguous Instructions**

User Instruction: Hi, I have an upcoming flight and I need to know how many suitcases I'm allowed to bring. I have a lot to pack, so I really need the correct information. Can you help me check this?

---

**Inconsistent Instructions**

User Instruction: Hi, I have an upcoming flight and I need to know exactly how many suitcases I'm allowed to bring with my reservation. I think I can bring either one or three suitcases, depending on the ticket, but I'm not sure. Can you confirm the correct number?

---

**Redundant Information**

User Instruction: Hi, I have an upcoming flight and I need to know exactly how many suitcases I'm allowed to bring with my reservation. I have a lot to pack, so I really need the correct number of suitcases I'm allowed to bring. Could you please tell me how many suitcases I can bring on this flight?

---

**Topic Drift**

User Instruction: Hi, I have an upcoming flight and I need to know exactly how many suitcases I'm allowed to bring with my reservation. By the way, do airlines usually serve meals on long flights, and are aisle seats better than window seats?

---

**Boundary Probing**

User Instruction: Hi, I have an upcoming flight and I need to know exactly how many suitcases I'm allowed to bring with my reservation. If I bring one extra suitcase beyond the allowed number, would it still be acceptable, or is there any way to avoid additional fees?

---

The following are examples of tool-side noise:

**Clean Tool Output**

FlightTool [SEP] Instruction: [SEP] Retrieve baggage allowance and reservation details for reservation ID WUNA5K.
[SEP] Execute.
Tool Response:

```
{
  "reservation_id": "WUNA5K",
  "origin": "ORD",
  "destination": "PHL",
  "cabin": "economy",
  "total_baggages": 1,
  "nonfree_baggages": 0
}
```

**Execution Failures (Service-Level Failure)**

FlightTool [SEP] Instruction: [SEP] Retrieve baggage allowance and reservation details for reservation ID WUNA5K.
[SEP] Execute.
Tool Response: ERROR: Service unavailable. The flight reservation system failed to respond (HTTP 503).

**Incomplete Responses (Partial Output)**

FlightTool [SEP] Instruction: [SEP] Retrieve baggage allowance and reservation details for reservation ID WUNA5K.
[SEP] Execute.
Tool Response:

```
{
  "reservation_id": "WUNA5K",
  "origin": "ORD",
  "destination": "PHL",
  "total_baggages": 1
```

**Erroneous Outputs (Incorrect Facts)**

FlightTool [SEP] Instruction: [SEP] Retrieve baggage allowance and reservation details for reservation ID WUNA5K.
[SEP] Execute.
Tool Response:

```
{
  "reservation_id": "WUNA5K",
  "cabin": "first_class",
  "total_baggages": 3,
  "nonfree_baggages": 0,
  "flight_number": "HAT999"
}
```

**Misleading Signals (Spurious Content)**

FlightTool [SEP] Instruction: [SEP] Retrieve baggage allowance and reservation details for reservation ID WUNA5K.
[SEP] Execute.
Tool Response:

```
{
  "reservation_id": "WUNA5K",
  "total_baggages": 1,
  "note": "Passengers on similar routes usually receive free upgrades
          and can often bring extra baggage without fees."
}
```

**Redundant Information (Low-Utility Content)**

FlightTool [SEP] Instruction: [SEP] Retrieve baggage allowance and reservation details for reservation ID WUNA5K.
[SEP] Execute.
Tool Response:

```
{
  "reservation_id": "WUNA5K",
  "user_id": "sophia_silva_7557",
  "origin": "ORD",
  "destination": "PHL",
  "flight_type": "round_trip",
  "cabin": "economy",
  "created_at": "2024-05-08T19:01:02",
  "created_at_timezone": "UTC",
  "created_at_unix": 1715194862,
  "system_log_id": "SYS-8839201",
  "debug_trace": "OK",
  "total_baggages": 1
}
```

# C. Agent Noise Robustness: Future Discussion

Although this study has preliminarily constructed an evaluation framework for agent noise robustness and revealed key phenomena, several critical paths remain to be explored to advance the field towards greater practicality and theoretical depth. The following outlines future research directions from four perspectives: refinement of noise modeling, innovation in training paradigms, expansion of the evaluation system, and theoretical understanding of robustness.

**(1) From Categorical Noise to Continuous Spectrum and Composite Noise Modeling** The current study categorizes noise into "user-side" and "tool-side" types, providing a clear initial analytical framework. However, real-world noise is far from binary. Future work should strive to construct a noise spectrum model with greater continuity and composability. For instance, user-side noise could extend from simple lexical perturbations (e.g., typos) to more complex phenomena like "intention drift" or "contextual contradictions in multi-turn dialogues." Tool-side noise should consider "temporal correlation" (e.g., intermittent failures) and "state dependency" (e.g., errors triggered only under specific input conditions). A more significant challenge lies in simulating the combination and synergistic effects of multiple noise sources—scenarios where ambiguous user instructions encounter delayed and incomplete tool responses. Building and evaluating such composite scenarios will better approximate the complexity of real-world deployments.

**(2) Paradigm Shift: From "Cleaned Training" to "Active Noise-Injected Training"** Most existing agent training relies on meticulously curated instructions and stable environments, creating a "fundamental mismatch" between training and deployment contexts. Drawing inspiration from data augmentation techniques in computer vision and reinforcement learning that enhance model robustness, future agent training must systematically incorporate noise exposure. This goes beyond merely adding random perturbations to data; it necessitates developing curriculum learning strategies. Initially, controlled, manageable noise is injected during training. As the agent's capabilities improve, the intensity, diversity, and unpredictability of the noise are gradually increased. Concurrently, adversarial training methods can be explored. By establishing a dynamic game between a "noise generator" and the agent, noise samples that effectively challenge the agent's current weaknesses can be generated, thereby targetedly enhancing the robustness of its vulnerable points.

**(3) From Static Benchmark Evaluation to Dynamic Simulation and Real User Interaction Evaluation** Agent-NoiseBench represents a significant step forward with static benchmarks, but the ultimate testing ground should be the dynamic, open real world. Future work must expand the evaluation system in two directions. First, developing high-fidelity, multimodal environment simulators, such as GUI operation simulators incorporating random network latency, rendering errors of interface elements, or unstructured document outputs, would make the injection of tool-execution noise more ecologically valid. Second, exploring online evaluation based on real user interactions is crucial. Through crowdsourcing platforms or pilot deployments, "organic noise"—produced by human users during natural interactions and not pre-programmable—can be collected. Such data is vital for understanding and modeling real user-side behavioral variations. This requires evaluation protocols capable of handling longer-term, more open-ended tasks and quantifying the agent's performance degradation and recovery capabilities during sustained interactions.

**(4) From Phenomenological Description to Theoretical Exploration of the Intrinsic Mechanisms of Robustness** The current study primarily documents the phenomenon of performance degradation and some correlative findings (e.g., the orthogonality between reasoning capability and noise robustness). Deeper scientific questions remain: How exactly does noise affect the internal computational processes of an agent? Future research needs to integrate interpretability analysis tools (e.g., neuron activation analysis, attention pattern tracing) to scrutinize the "breaking points" in the agent's reasoning chain under noise interference. For example, does tool noise primarily disrupt the maintenance of working memory? Does user noise interfere with the parsing of initial conditions for task planning? Understanding these mechanisms is key to designing more targeted interventions. Furthermore, a more solid theoretical framework is needed to formally define agent robustness. This may involve borrowing concepts like stability from control theory or channel capacity against interference from information theory, providing tools for analyzing the upper and lower bounds of robustness for agents with different architectures and training methodologies.

Research on agent noise robustness should not stop at building a more challenging "stress test" benchmark. Its long-term goal is to steer the entire field from pursuing "optimal lab performance" towards constructing "trustworthy agents for the real world." This demands that we not only evaluate agents in noise but also train agents in noise and understand agents through theory, ultimately achieving a robust symbiosis between agents and the imperfect, uncertain real world. This study is merely the starting point of this journey, and we anticipate more researchers will join this interdisciplinary area, which holds both significant practical value and profound theoretical significance.

