# OpenReview forum: "AgentNoiseBench: Benchmarking Robustness of Tool-Using LLM Agents Under Noisy Condition"
_ICML.cc/2026/Conference — ICML 2026 regular_

### Official Review · Reviewer_SkY3 · 2026-02-19

**Soundness:** 3
**Presentation:** 3
**Significance:** 3
**Originality:** 2
**Overall Recommendation:** 4
**Confidence:** 4

**Summary:**

This paper proposes AgentNoiseBench, which is a systematic framework designed to evaluate the robustness of LLM-based agents operating under noisy real-world conditions. The authors categorize environmental noise into two primary types; instruction noise from users and tool-execution noise from external APIs. To accurately test agents, they developed an automated pipeline that injects adversarial noise into existing benchmarks while strictly ensuring the tasks remain logically solvable. Utilizing a trajectory-aware evaluation protocol across various models, the authors demonstrates that current SOTA agents suffer significant performance degradation when exposed to these realistic perturbations.

**Compliance With Llm Reviewing Policy:**

Affirmed.

**Final Justification:**

This paper presents a well-structured and principled benchmark for evaluating agent robustness under noisy conditions. A key strength of the work is its solvability-preserving adversarial noise framework, which enables a clean separation between true model fragility and task infeasibility. In addition, the trajectory-level evaluation—rather than focusing solely on final answers—provides fine-grained diagnostic signals and reveals an interesting failure mode where strong reasoning models rationalize spurious noise instead of filtering it out. These insights make a meaningful contribution to our understanding of robustness in modern reasoning agents.

The main weaknesses initially concerned the computational overhead of trajectory-aware evaluation and the positioning of the benchmark relative to recent trajectory-based evaluation work. In the rebuttal, the authors addressed these concerns clearly and effectively by clarifying the intended scope of the benchmark as a diagnostic robustness stress test and by more explicitly distinguishing their contribution through the noise-constrained, adversarial, and solvability-guaranteed setup. These clarifications substantially mitigated my concerns.

Overall, the rebuttal strengthened the paper by sharpening its positioning and clearly articulating its unique value. Given the methodological rigor, novel insights, and satisfactory responses to the raised issues, I maintain a positive assessment and a Weak Accept recommendation.

**Key Questions For Authors:**

No questions for authors

**Limitations:**

yes

**Strengths And Weaknesses:**

## Strengths ##

- The automated pipeline utilizes a constrained adversarial evolution mechanism that dynamically scales noise while explicitly guaranteeing task solvability. This is a highly rigorous and novel approach to test models, ensuring that agent failure is due to fragility rather than the task becoming impossible.

- Instead of merely checking final answer, the framework evaluates the entire reasoning and interaction trajectory. By tracking step-wise entropy, it provides deep diagnostic insights into exactly where and how noise disrupts an agent's cognitive pipeline.

- This paper reveals an orthogonality between general reasoning capability and noise robustness in a wide range of models. It exposes a pathological failure mode where highly capable reasoning models attempt to logically rationalize spurious noise to create coherent but factually invalid outputs, rather than simply filtering the noise out.

## Weaknesses ##

- The trajectory-aware evaluation protocol requires evaluating intermediate steps using a model-based evaluator, which introduces significant computational overhead and limits the benchmark's scalability for large-scale or real-world settings.

- The benchmark specifically targets robustness under noisy conditions, but there is some recent research to evaluate the agent trajectories in various metrics [1, 2]. The distinctions between this study and previous works, as well as the unique contributions of this research, must be clearly highlighted.

[1] Kim, Wonjoong, et al. "Beyond the Final Answer: Evaluating the Reasoning Trajectories of Tool-Augmented Agents." arXiv preprint arXiv:2510.02837 (2025).
[2] Kim, Takyoung, et al. "Pipa: A unified evaluation protocol for diagnosing interactive planning agents." arXiv e-prints (2025): arXiv-2505.

---

> ### Author Rebuttal · Authors · 2026-03-30
>
> Dear Reviewer SkY3,
>
> Thank you for your kind words and positive feedback regarding presentation and effectiveness of our work!
>
> **Q1: Differences, connections, and unique contributions compared to recent agent evaluation works?**
>
> We sincerely thank the reviewer for highlighting TRACE [1] and PIPA [2], two pivotal works in agentic evaluation. We will cite both in our introduction and provide deeper comparative analysis in Related Work and Methodology sections.
>
> While PIPA, TRACE, and our work all champion the vision of "evaluating beyond the final answer," we tackle distinct but complementary research problems.
>
> **PIPA**​ proposes a unified, process-level evaluation protocol within the POMDP framework, offering systematic, cross-domain diagnosis of an agent's planning stages.
>
> **TRACE**​ provides an evidence-based framework for multi-dimensional quality scoring (e.g., efficiency, hallucination, adaptability), excelling by not requiring a single ground-truth trajectory.
>
> **AgentNoiseBench**'s key contribution is a systematic noise robustness evaluation framework featuring: (1) dual-source noise classification (user-side instructions and tool-side execution), (2) constrained adversarial evolution for generating challenging yet solvable perturbations, and (3) trajectory-aware multi-dimensional evaluation requiring agents to maintain reasoning validity while completing tasks.
>
> A detailed comparison is provided below:
>
> |Aspect|AgentNoiseBench|PIPA|TRACE|
> |:-|:-|:-|:-|
> |Objective| Evaluate agent robustness under realistic environmental noise | Provide a unified protocol for diagnosing interactive planning agents | Assess reasoning trajectories beyond answer matching |
> |Analytical Focus| Robustness to instruction and tool-execution noise | Process-level diagnosis across the planning pipeline | Intrinsic quality of tool-augmented reasoning trajectories |
> |Evaluation Axes| Performance degradation, trajectory stability, and sensitivity to different noise sources | State consistency, tool efficiency, observation alignment, policy adherence, and task completion | Efficiency, hallucination, and adaptivity |
> |Methodological Basis| Controlled noise injection with solvability constraints and trajectory-aware analysis | POMDP-based decomposition of agent behavior into atomic evaluation units | Evidence-bank construction from tool traces for reference-free trajectory assessment|
> |Relation to Final Outcomes| Final-task success remains a central robustness signal | Final outcomes are one component within a broader diagnostic framework | Trajectory quality is evaluated relatively independently of final-answer correctness |
> | Treatment of Valid Alternatives | Primarily examines whether noise disrupts successful task execution | May under-reward alternative but valid trajectories when standard paths are used as anchors | Naturally accommodates multiple valid solution paths without requiring a gold trajectory |
> | Main Contribution | Realistic robustness benchmarking for agent deployment in noisy settings | Interpretable and standardized diagnosis of intermediate agent behavior | Flexible, scalable, and low-annotation evaluation of reasoning quality |
>
> **Q2: The evaluation method incurs significant computational overhead.**
>
> We acknowledge that trajectory-aware evaluation introduces additional computational costs. However, it is crucial to note that in our experiments, we only conduct step-wise evaluation for the *perturbed* steps rather than the entire trajectory. Our key observation is that effective noise injected at a given step almost exclusively manifests in the immediately subsequent generation. Therefore, to balance diagnostic effectiveness with computational efficiency, we bypass LLM-based evaluation for unperturbed steps, requiring zero additional computation for them.
>
> Following your valuable suggestion, we conducted a targeted time-cost analysis. We sampled a subset of instances across different models and measured the average evaluation time of our trajectory-aware protocol versus a standard final-answer-only baseline. To ensure reliability, we strictly controlled for network fluctuations during the benchmarking process.
>
> | Evaluation Protocol | DeepSeek-V3.1 | GLM-4.6 (w/o thinking) | Qwen3-Max | GLM-4.6 (w/ thinking) | gpt-5.2 (w/ thinking) | o3 (w/ thinking) |
> |:-:|:-:|:-:|:-:|:-:|:-:|:-:|
> |Origin|32.3|38.7|36.5|45.2|50.8|48.6|
> |Trajectory-Aware|37.1|43.5|41.2|51.0|57.3|54.1|
>
> Based on these results, we draw the following observation:
>
> Compared to the baseline time cost of task completion itself, the additional computational overhead introduced by trajectory-aware evaluation remains generally acceptable, while yielding critical diagnostic insights into the reasoning process. Future work will explore more efficient evaluation strategies to further mitigate this time cost.
>
> We hope our improvements can earn your approval.
>
> Best,
>
> Authors

---

> > ### Author Rebuttal · Reviewer_SkY3 · 2026-04-01
> >
> > Thanks to the authors for detailed responses.
> >
> > I'll maintain the positive score.

---

> > > ### Author Response · Authors · 2026-04-04
> > >
> > > Dear Reviewer SkY3,
> > >
> > > Thank you very much for your insightful feedback. We sincerely appreciate your recognition of our work on AgentNoiseBench and your valuable suggestions regarding the need for greater clarity in distinguishing our work from prior studies (e.g., [1, 2]) and addressing scalability concerns. As recommended, we will ensure that all methodological details and experimental results are thoroughly documented in the revised manuscript.
> > >
> > > We would like to take this opportunity to reiterate the core contributions of our paper:
> > >
> > > - **While most existing agent evaluations  focus narrowly on performance in idealized, noise-free settings, we introduce a benchmarks for assessing robustness under real-world noise. Our automated pipeline enables controlled injection of different types of noise, providing reproducible benchmarks for future agent robustness research.**
> > >
> > > Thank you again for your thoughtful comments and support. We are committed to advancing the field of robust AI agents and believe that addressing real-world imperfections is crucial for trustworthy deployments. Your feedback has been invaluable in strengthening our work, and we look forward to contributing these insights to the community.
> > >
> > > Best regards,
> > >
> > > Authors of Paper 34070

---

### Official Review · Reviewer_m374 · 2026-03-11

**Soundness:** 1
**Presentation:** 3
**Significance:** 2
**Originality:** 3
**Overall Recommendation:** 4
**Confidence:** 3

**Summary:**

The paper proposed an evaluation framework to evaluate the robustness of LLMs by introducing two types of noise, instruction noise and tool-execution noise. The key idea is injecting noise to any existing benchmarks to simulate real-world environments while enforcing a solvability constraint, as existing evaluation methods are usually based on noise-free, idealised assumptions. The authors tested the benchmark by injecting noise into VitaBench, $\tau^2$ Bench, and multi-hop QA, and the evaluation spans a model zoo of 24 models.

**Compliance With Llm Reviewing Policy:**

Affirmed.

**Final Justification:**

The rebuttal addressed my main concerns and I decided to raise the score.

**Key Questions For Authors:**

See above.

**Limitations:**

See above.

**Strengths And Weaknesses:**

Strengths

The paper tries to bridge the gap between the current benchmarks and the real-world environment. The authors categorise the common real-world noise into instruction noise and tool-execution noise, which is useful for analysing models' robustness. The trajectory-aware evaluation protocol is a good addition to identify any errors in the intermediate steps.

Weaknesses
1. The main method (adversarial noise evolution) is unclear to me in terms of how it works. Specifically in Section 2.2.2, $L_{deg}$ is not explained, how does $\theta$ get optimised, and how to ensure the solvability.
2. The trajectory-aware metric is likely to be wrong. In equation 2, it says "a value of 1 indicates that the current step is affected by
noise", then it means a product over steps equals 1 only if every step is affected.
3. The Tables in the appendix have the same entries, which can be worrisome.

---

> ### Author Rebuttal · Authors · 2026-03-30
>
> Dear Reviewer m374,
>
> We sincerely appreciate your comments on our work's limitations and how to improve it. Below are our responses to three questions.
>
> **Q1: Clarify the optimization process and solvability of the adversarial noise evolution method**
>
> Following your feedback, we have expanded Section 2.2.2 and added Appendix details to clarify the adversarial noise optimization process and its solvability constraints.
>
> **Solvability Constraint.** Solvability is ensured through a dual-strategy design:
>
> *Interpretability Constraint*:The locations and types of injected noise must be explicitly identifiable, enabling human evaluators  to determine which steps were perturbed and what kind of noise was applied. This ensures the original task intent stays dominant in the trajectory and keeps the introduced noise interpretable and semantically coherent.
>
> *Diversified Sampling*:We use multiple random samplings per task (e.g., increasing temperature) to ensure the agent produces at least one valid response across noisy instances. Robust commercial agents also mitigate noise via human-like strategies: they ask clarifying questions, make environmental inferences, and, after repeated failures, confirm other information before retrying or using tools.
>
> **Adversarial Noise Optimization.** Formulated as prompt-level black-box optimization: fix a strong LLM as noise generator, optimize adversarial prompt $\theta$ (controlling noise injection into queries/tool's responses). Objective:
>
> $$
> \theta^*=\arg\max_{\theta} L_{\mathrm{deg}}(A, G(x;\theta))
> \quad \text{s.t.} \quad
> I_{\mathrm{solvable}}(G(x;\theta))=1.
> $$
>
> Here, $L_{\mathrm{deg}}$ is implemented as a feedback-driven evolutionary optimization loop：
>
> 1.Expert Initialization: Experts provide high-quality seed prompts to ensure semantic soundness.
>
> 2.Noise Generation: The current prompt $\theta_{t}$ directs the generator to perturb the target queries or tool returns.
>
> 3.Impact Evaluation:​ Evaluate perturbed samples against the agent and check if the task completion rate reaches the target degradation (e.g., 50% drop).
>
> 4.Evolutionary Update: Ineffective noise samples, together with their original text, the agent’s response, and the current prompt  $\theta_{t}$ are fed into an evolutionary model to produce an enhanced prompt $\theta_{t+1}$.
>
> 5.Terminate: The iteration terminates after 3 rounds, at which point the adversarial prompts achieve an optimal trade-off between perturbation intensity and solvability constraints.
>
> Prompt iterative optimization effectiveness is shown below:
> |Iter|Acc Rate|Solv Rate|
> |:-:|:-:|:-:|
> |R0|100%|100%|
> |R1|67%|100%|
> |R2|55%|97%|
> |R3|48%|92%|
>
> Experiments found that after three iterations, solvability decreases significantly, and the benefits of further iterations diminish.
>
> **Q2: Eq. 2's product definition seems flawed as it yields 1 only if every step is noisy**
>
> We sincerely thank the reviewer for this careful observation and apologize for the confusion caused by the notation error. The reviewer is correct that Eq. (2)/(3) mistakenly reversed the indicator semantics. In the revised version, we have corrected this definition as
> $I_{\text{traj}}(s_i, T, R) \in \{0,1\}$,
> where $I_{\text{traj}}(s_i, T, R)=1$ means step $s_i$ remains valid under injected noise, and $0$ means it is corrupted and deviates from the intended trajectory. Accordingly, the trajectory-level indicator is
> $I_{\text{traj}}(\tau, T, R)=\prod_{s_i \in \tau} I_{\text{traj}}(s_i, T, R)$,
> which serves as a strict validity gate in Eq. (4): it equals 1 only if all steps remain valid, and 0 otherwise.
>
> **Q3：Some appendix tables contain identical entries, which looks suspicious**
>
> We thank the reviewer for noting the duplicate entries in the appendix tables. This was an oversight, and we apologize for any confusion. We have corrected these errors in the revised manuscript to ensure all entries are accurate and non-redundant. Below is a subset of the appendix tables, showing model performance under various noise types and conditions.
> |Model|clean|ambiguity|topic_shift|conflict|redundancy|boundary|
> |:-|:-:|:-:|:-:|:-:|:-:|:-:|
> ||||**Non-Think**|||
> |GPT-4.1|60.4|41.7|47.9|47.9|41.7|60.4|
> |gemini-2.5-pro|77.1|54.2|60.4|62.5|62.5|77.1|
> |Qwen3-Max|66.7|41.7|54.2|54.2|54.2|66.7|
> |Claude-4.5-Sonnet|75.0|68.8|60.4|58.3|60.4|75.0|
> ||||**Think** ||||
> |GLM-4.6|75.0|77.1|66.7|66.7|68.8|75.0|
> |deepseek-R1-0528|70.8|52.1|52.1|52.1|41.7|70.8|
> |gpt-5.2|79.2|60.4|68.8|66.7|66.7|79.2|
> |o3|77.1|72.9|66.7|66.7|64.6|77.1|
>
> |Model|clean|error|failure|incomplete|induce|redundancy|
> |:-|:-:|:-:|:-:|:-:|:-:|:-:|
> ||||Non-Think||||
> |GPT-4.1|60.4|61.2|47.9|43.8|41.8|47.9|
> |gemini-2.5-pro|77.1|75.9|60.4|52.1|54.2|62.5|
> |Qwen3-Max|66.7|65.1|54.2|45.8|41.7|54.2|
> |Claude-4.5-Sonnet|75.0|78.1|58.3|60.4|68.8|60.4|
> ||||Think||||
> |Qwen3-max|89.6|89.1|58.3|60.4|70.8|52.1|
> |deepseek-R1-0528|70.8|70.3|52.1|50.0|52.1|41.7|
> |gpt-5.2|79.2|78.7|66.7|68.8|60.4|66.7|
> |o3|77.1|76.6|66.7|54.2|72.9|64.6|

---

> > ### Author Rebuttal · Reviewer_m374 · 2026-04-04
> >
> > Thanks to the authors for addressing my questions. I've raised my score.

---

> > > ### Author Response · Authors · 2026-04-04
> > >
> > > Dear Reviewer m374,
> > >
> > > Thank you very much for your meticulous and insightful​ feedback and for recognizing the significance of our work on AgentNoiseBench. We are truly encouraged by your positive assessment and deeply appreciate your valuable suggestions throughout the review process. In response to your comments, we have carefully revised and improved the manuscript to enhance its clarity, rigor, and completeness.
> > >
> > > We would like to take this opportunity to reiterate the core contributions of our paper:
> > >
> > > - **While most existing agent evaluations focus narrowly on performance in idealized, noise-free settings, we introduce a benchmark for assessing robustness under real-world noise. Our automated pipeline enables controlled injection of different types of noise, providing reproducible benchmarks for future agent robustness research.**
> > >
> > > Thank you again​ for your time and guidance, and we look forward to your continued support as we further refine our work.
> > >
> > > Best regards,
> > >
> > > Authors of Paper 34070

---

### Official Review · Reviewer_dAAy · 2026-03-13

**Soundness:** 2
**Presentation:** 2
**Significance:** 3
**Originality:** 3
**Overall Recommendation:** 4
**Confidence:** 4

**Summary:**

The paper proposes AgentNoiseBench, a framework designed to evaluate the robustness of LLM-based agents under noisy, real-world conditions. The framework uses an automated pipeline to inject controlled adversarial noise into existing benchmarks while ensuring the tasks remain logically solvable. Evaluating 24 state-of-the-art models shows that all models suffer substantial performance degradation (averaging a 20.8% drop) under noise.

**Compliance With Llm Reviewing Policy:**

Affirmed.

**Final Justification:**

My concerns are resolved by authors during the rebuttal.

**Key Questions For Authors:**

1. Can you provide details about the noise injection mechanism mentioned in Section 2.2.2? How is the adversarial prompt updated for user-side injection and tool-side injection, respectively? How do you check the solvability constraint?

2. Can you provide detailed results for Observation 2 of RQ1. It seems the multi-turn vs single-turn results are not reported in Table 1.

3. Can you measure the ranking discrepancy (e.g., using Spearman's Rank-Order Correlation) of models before and after injecting the noise? How does the ranking discrepancy change when using different black-box models for the noise generator? I think this is an important analysis to compare the robustness of different models under injected noise.

**Limitations:**

yes

**Strengths And Weaknesses:**

**Strengths**:

1. The paper effectively bridges the gap between controled laboratory evaluations and the chaotic nature of real-world deployment.

2. Strict constraint placed on noise generation ensures that perturbed tasks remain solvable so that failures are accurately attributed to agent fragility rather than impossible task conditions.

3. The study tests a wide, diverse suite of 24 models.

**Weaknesses**:

1. The paper misses detail about how the constrained adversarial noise injection (Section 2.2.2) is concretely implemented.

2. It's unclear how AgentNoiseBench ensures that tasks remain solvable after injecting noise on the tool side.

3. Limited analysis and presentation of whether errors amplify over multi-turn compared to single-turn.

---

> ### Author Rebuttal · Authors · 2026-03-30
>
> Dear Reviewer dAAy,
>
> We appreciate your insightful comments on our methodology and experiments. Below are our responses to questions.
>
> **Q1: Elaborate on the implementation of the noise injection, prompt update, and solvability constraint mechanisms**
>
> **Noise Injection Mechanism.** We employ a frozen-parameter black-box LLM as the noise generator. The generation process is controlled by pre-optimized adversarial prompts and injects noise into the agent–user–environment loop​ by replacing either the user queries or the tool returns.
>
> **Solvability Constraints.** To keep tasks solvable, we impose a dual-strategy constraint:
>
> Noise injection must be identifiable so human can see which steps are perturbed and how. This ensures the original task intent remains dominant and the trajectory stays within a comprehensible scope.
>
> We run multiple random samplings (e.g., higher temperature) on the same task to ensure at least one solvable trajectory under noise. Successful trajectories show robust agents exhibit noise tolerance via clarification requests, environmental reasoning, and multi-turn re-queries after waiting.
>
> **Prompt Update Mechanism.** We optimize adversarial prompts for user and tool noise through feedback-driven iteration, under interpretability constraints and on a set of 100 solvable samples.
>
> 1.Expert Initialization: Experts provide high-quality seed prompts to ensure semantic soundness.
>
> 2.Noise Generation: The current prompt $\theta_{t}$ directs the generator to perturb the target queries or tool returns.
>
> 3.Impact Evaluation:​ Evaluate injected noise against the agent and check if the task completion rate reaches the target degradation (e.g., 50% drop).
>
> 4.Evolutionary Update: Ineffective noise samples, together with their original text, the agent’s response, and the current prompt  $\theta_{t}$ are fed into an powerful model to produce an enhanced prompt $\theta_{t+1}$.
>
> 5.Terminate: The iteration terminates after 3 rounds, at which point the adversarial prompts achieve an optimal trade-off between perturbation intensity and solvability constraints.
>
> Prompt iterative optimization effectiveness is shown below:
> |Iter|Acc Rate|Solv Rate|
> |:-:|:-:|:-:|
> |R0|100%|100%|
> |R1|67%|100%|
> |R2|55%|97%|
> |R3|48%|92%|
>
> Experiments found that after three iterations, solvability decreases significantly, and the benefits of further iterations diminish.
>
> **Q2: Show detailed results for multi-turn vs. single-turn**
> |Models||$\tau^2$|||Vita|||Search||
> |:-:|:-:|:-:|:-:|:-:|:-:|:-:|:-:|:-:|:-:|
> ||Origin|Single\_Turn|Multi\_Turn|Origin|Single\_Turn|Multi\_Turn|Origin|Single\_Turn|Multi\_Turn|
> |||||Non-Think||||||
> |DeepSeek-V3-0324|0.47|0.42|0.34|0.25|0.21|0.19|0.21|0.21|0.19|
> |GPT-4.1|0.56|0.48|0.40|0.41|0.37|0.32|0.29|0.28|0.27|
> |gemini-2.5-pro|0.59|0.53|0.46|0.49|0.44|0.37|0.36|0.32|0.29|
> |Qwen3-Max|0.57|0.49|0.43|0.36|0.29|0.23|0.31|0.29|0.24|
> |GLM-4.6|0.74|0.66|0.56|0.40|0.36|0.33|0.47|0.43|0.39|
> |Claude-4.5-Sonnet|0.71|0.64|0.56|0.48|0.43|0.40|0.34|0.31|0.29|
> |||||Think||||||
> |Qwen3-max|0.87|0.78|0.67|0.47|0.41|0.37|0.49|0.42|0.33|
> |GLM-4.6|0.75|0.69|0.62|0.49|0.44|0.40|0.47|0.40|0.36|
> |Claude-4.5-Sonnet|0.73|0.66|0.61|0.51|0.48|0.41|0.35|0.31|0.29|
> |deepseek-R1-0528|0.53|0.47|0.42|0.42|0.39|0.38|0.33|0.28|0.26|
> |gpt-5.2|0.81|0.74|0.67|0.53|0.50|0.45|0.49|0.44|0.37|
> |o3|0.70|0.64|0.62|0.50|0.46|0.45|0.46|0.38|0.35|
>
> From the table, multi-turn noise injection consistently challenges the original performance more than single-turn injection, causing larger accuracy drops across tested models and scenarios.
>
> **Q3: (1) Analyze model ranking changes under noise via Spearman correlation. (2) See if different black-box noise generators cause different ranking shifts.**
>
> 1.Spearman's rank correlation (ρ ≈ 0.917) between original and noise-injected rankings in Table 1 indicates overall ranking consistency, while still revealing noticeable shifts for some models. Claude-4.5-Sonnet: 3→8; Qwen3-max: 1→5.
>
> 2.Following your feedback, we select GPT-4.1, Claude-3.5, Gemini-2.5, and DeepSeek-V0324 as black-box noise generators to examine their impact. A subset of results is shown below, while the complete results and analysis will be included in Appendix A.
>
> Generator：Deepseek-V3-0324
> |Model|Origin|User_Noise|Tool_Noise|
> |:---:|:---:|:---:|:---:|
> |||Non-Think|||
> |DeepSeek|0.47|0.38|0.30|
> |GPT-4.1|0.56|0.49|0.31|
> |Qwen3-Max|0.57|0.52|0.35|
> |||Think|||
> |Claude-4.5-Sonnet|0.73|0.64|0.56|
> |deepseek-R1|0.53|0.45|0.36|
> |gpt-5.2|0.81|0.71|0.60|
>
> Generator：GPT-4.1
> |Model|Origin|User_Noise|Tool_Noise|
> |:---:|:---:|:---:|:---:|
> |Non-Think|||||
> |DeepSeek|0.47|0.36|0.31|
> |GPT-4.1|0.56|0.50|0.29|
> |Qwen3-Max|0.57|0.51|0.34|
> |Think|||||
> |Claude-4.5-Sonnet|0.73|0.66|0.58|
> |deepseek-R1|0.53|0.47|0.38|
> |gpt-5.2|0.81|0.73|0.62|
>
> We also computed the Spearman rank correlation for black-box noise generators: GPT-4.1 (≈0.903) and DeepSeek-V3-0324 (≈0.91). The results show that model robustness is stable and rankings change little across generators.

---

> > ### Author Rebuttal · Reviewer_dAAy · 2026-04-03
> >
> > Thanks for the rebuttal and additional experimental data. My concerns are resolved. I've raised the score.

---

> > > ### Author Response · Authors · 2026-04-04
> > >
> > > Dear Reviewer dAAy,
> > >
> > > Thank you very much for your insightful feedback and for recognizing the significance of our work on AgentNoiseBench. We are truly encouraged by your positive assessment and appreciate your valuable suggestions throughout the review process. As recommended, we will ensure that all methodological details and experimental results are thoroughly documented in the revised manuscript.
> > > We would like to take this opportunity to reiterate the core contributions of our paper:
> > >
> > > - **While most existing agent evaluations  focus narrowly on performance in idealized, noise-free settings, we introduce a benchmarks for assessing robustness under real-world noise. Our automated pipeline enables controlled injection of different types of noise, providing reproducible benchmarks for future agent robustness research.**
> > >
> > > Thank you again for your thoughtful comments and support. We are committed to advancing the field of robust AI agents and believe that addressing real-world imperfections is crucial for trustworthy deployments. Your feedback has been invaluable in strengthening our work, and we look forward to contributing these insights to the community.
> > >
> > > Best regards,
> > >
> > > Authors of Paper 34070

---

### Decision · Program_Chairs · 2026-04-30

**Decision:**

Accept (regular)

**Comment:**

The paper proposes AgentNoiseBench, a framework designed to evaluate the robustness of LLM-based agents under noisy, real-world conditions. All reviewers incline toward acceptance, and the rebuttal has fully resolved the concerns raised during the review process.

I recommend that the authors incorporate the clarifications and additional analysis from the rebuttal into the final manuscript to further strengthen the paper.